# Mapping of HOCl-oxidized RNA identifies abasic sites as major damage and oxidation product of oxo$^8$G

Marlies Weber[1,5], Kasturi Raorane[2,5], Clara Johanna Grampp[1], Valérie Bourguignon[2,3], Lea-Marie Kilz[1], David Glänzer[4], Virginie Marchand [3], Christoph Kreutz [4], Yuri Motorin [2,3] ✉ & Mark Helm [1] ✉

RNA oxidation is an important yet understudied process, partly because methods to localize oxidized residues in RNA are lacking. We introduce OAbSeq, a deep-sequencing approach that maps oxidized sites with high sensitivity by exploiting aniline-induced strand scission at noncanonical nucleosides to generate unique ligation-competent fragments utilized for library preparation. Applied to yeast RNA, OAbSeq detects widespread signals predominating at purines, especially at guanosines. Exogenous oxidation increased signal intensity but preserved the guanosine-dominated pattern. Parallel quantification of 8-oxoguanosine (oxo$^8$G) and abasic sites revealed that abasic sites are more abundant than oxo$^8$G following oxidative treatment in vitro and under physiological conditions. These data support a model in which guanosine oxidation proceeds via transient oxo$^8$G yielding abasic sites that can be mapped at nucleotide resolution by OAbSeq. Our findings also suggest abasic sites may be a more informative marker of RNA oxidative damage than oxo$^8$G, facilitating studies of RNA oxidation dynamics in cells.

Mapping of noncanonical nucleoside structures, i.e., their detection within an RNA sequence, is a field of current high interest with roots at the very basics of chemical direct RNA sequencing as developed by Peattie & Gilbert[1,2], itself going back to the same style of DNA sequencing developed by Maxam & Gilbert[3]. Much renewed interest in the underlying chemistry comes from the possibility of combining it with high throughput sequencing of cDNA, e.g. using Illumina technology[4–6]. Many current RNA modification detection methods rely on chemical treatments, themselves tailored to specific RNA modifications. Typical outcomes of such treatments are specific RNA cleavage at (or adjacent to) the modified residue, or alterations of the so-called reverse transcription signature (RT-signature), which embodies the transfer of site-specific and chemistry-specific information from RNA into cDNA[4–7]. The vast majority of published mapping

methods focuses on RNA modifications, typically defined as post-transcriptional alterations of nucleotide chemical structure, performed by enzymes with a certain measure of site specificity. In a few cases, however, the principles have also been applied to chemical alterations of RNA that occur at very low stoichiometry and with wide and unfocused distributions. One such group of methods derives from structural probing, for example, with alkylating agents[2,8], another one focuses on the distribution of oxidized nucleobases in transcriptomes[9,10]. The latter is based on a specific RT-signature, which includes the incorporation of dATP into cDNA opposite 8-oxoguanosine (oxo$^8$G, see Supplementary Tables 1 and 2 for abbreviations)[11], the best explored ribonucleoside damage so far. In addition to oxo$^8$G, a number of ring-opened guanosine oxidation products are known[12–16]. Oxidation products of adenine, cytosine and

[1]Johannes Gutenberg-University Mainz, Institute of Pharmacy and Biomedical Sciences, Staudingerweg, Mainz, Germany. [2]Université de Lorraine CNRS, IMoPA UMR7365, Nancy, France. [3]Université de Lorraine CNRS, INSERM, UAR2008 IBSLor Epitranscriptomics and RNA Sequencing Core Facility, Nancy, France. [4]Institute of Organic Chemistry and Center for Molecular Biosciences Innsbruck (CMBI), University of Innsbruck, Innrain 80/82 6020, Innsbruck, Austria. [5]These authors contributed equally: Marlies Weber, Kasturi Raorane. ✉e-mail: yuri.motorin@univ-lorraine.fr; mhelm@uni-mainz.de

uridine have been less extensively investigated, and include 8-oxoadenine (8-oxoA) and the 5-hydroxypyrimidines[15–23]. Oxidized ribonucleotides in general are not being regarded as bona fide RNA modifications, as they occur as byproducts and indirect consequences of spontaneous chemical reactions unrelated to enzymatic RNA metabolism. Therefore, such oxidative modifications are not site-specific, even if some preferential nucleotide environments can be found[24–28].

The arguably most important source of RNA oxidation is reactive oxygen species (ROS), generated as byproducts of cellular respiration in mitochondria or enzymatically catalyzed redox reactions, to name two important ones[29–32]. Rather than as byproducts, oxidants like superoxide anions[30,33] and hypochlorous acid (HOCl) are known to be generated by dedicated enzymes to combat pathogens. Immune cells such as neutrophils express the enzyme myeloperoxidase (MPO), which in turn catalyzes the formation of HOCl by utilizing hydrogen peroxide and chloride ions[34–36]. HOCl is a potent oxidant and a weak acid ($pK_a = 7.46$) leading to chlorination and oxygenation of various biomolecules[34,35,37]. In nucleic acids this includes formation of 8-oxopurines, 8-chloropurines as well as 5-hydroxy- and 5-chloropyrimidines[34,35,38–41]. Recognition of these oxidized nucleosides facilitates inflammatory processes[30,37,42] and induces alterations in transcription and translation[22,43–45].

Safe for their distribution and stoichiometry, there is no fundamental difference between genuine RNA modification and spontaneously formed RNA oxidation products from an analytical perspective. Both types are, e.g., amenable to quantification by LC-MS[46–50]. Also, similar to RNA modifications, certain oxidized nucleotide species are amenable to aniline cleavage of the type used by Peattie for direct RNA sequencing[1]. Aniline cleavage is conducted by treatment with an acidic aniline solution, concomitant with heating. The latter leads to a β-elimination reaction of an iminium species formed in the former step. The Burrows lab has analyzed aniline cleavage rates for a number of RNA nucleobase oxidation products, finding certain of them susceptible[51]. Interestingly, oxo[8]G, the most prominent product of RNA oxidation, was among the least reactive species investigated, while some of the higher guanosine oxidation products underwent cleavage with considerable efficacy. For aniline cleavage to occur, the nucleobase or its surrogate must act as a leaving group, while aniline acts as nucleophile, yielding said iminium ion. However, it has not been conclusively shown if the leaving group is directly substituted by aniline, or if the reaction commences by nucleobase leaving and proceeds via an intermediate abasic site. There is consensus, however, that an abasic site will readily undergo aniline cleavage[52–55]. Aniline cleavage is the basis for many analytical sequencing approaches[1,56–58] and, importantly, results in a 5′-phosphorylated fragment downstream of the cleavage site. This molecular detail is of central importance to a mapping method called AlkAnilineSeq (AAS)[56]. AAS is a particularly sensitive, semi-quantitative method for mapping certain noncanonical nucleotides in RNA (Fig. 1a). The first chemical treatment is a brief heating at alkaline pH, which also fragments the RNA to a size compatible with Illumina sequencing. Following this chemical treatment, RNA is enzymatically dephosphorylated by phosphatase to remove all 5′-phosphates, such that all thereafter newly generated 5′-phosphates are specific to aniline cleavage sites. This step also removes all 3′-phosphates (including 2′,3′-cyclophosphates) resulting from initial alkaline fragmentation, making RNA fragments compatible with further adapter ligation. The next step is a classical aniline scission treatment, and the resulting 5′-phosphates constitute specific and therefore privileged entry points for the ensuing small RNAseq library preparation. Thus, the specificity of AAS derives from an adapter ligation step that requires an RNA fragment with a 5′-phosphate, such as one derived from aniline cleavage, which leads to high signal strength, signal/noise ratio, and sensitivity.

During the development of AAS, we analyzed the effect of various permutations of the above steps and found that the alkaline treatment made certain modified RNA nucleosides amenable to detection by subsequent aniline scission. This is remarkable, since, e.g., in the case of m[7]G and m[3]C, the original protocols by Peattie & Gilbert require additional treatments by NaBH$_4$ and hydrazine, respectively, for efficient aniline cleavage[1,2]. We take these findings as an indication that the alkaline treatment does indeed create aniline-susceptible fragilized nucleotides, since we found that its omission or replacement by RNA fragmentation with magnesium ions at high temperatures (but neutral pH) almost totally ablated cleavage signals[59,60]. Originally established for m[3]C and m[7]G, AAS has meanwhile been shown to also detect D and ho[5]C[56]. The corresponding structures suggest that an overarching property of reactive nucleotide structures might be a fragile glycosidic bond, caused by nonaromatic structures or nucleobase derivatives with improved leaving group properties. The most plausible explanation is that during alkaline treatment, a hydroxide anion displaces the noncanonical nucleobase, resulting in an abasic site. However, depending on the exact structure of the nucleobase, the latter may also directly react with a hydroxide anion, thereby being transformed into an improved leaving group.

In the backdrop of the above, we here pursue the recurrent observation of substoichiometric AAS signals at purine sites, found in all RNAs investigated so far. We used hypochlorous acid (HOCl) as a chemical tool to elucidate the molecular genesis of signals in living yeast. Yeast was employed as a model organism that does not produce HOCl, and we could therefore demonstrate the oxidative action of HOCl on RNA in living cells via LC-MS-based quantification of chlorinated ribonucleosides. We show that purine signals originate from RNA oxidation, but that the contribution of the most prominent RNA oxidation product oxo[8]G to such signals is negligible. Instead, we identified abasic sites as the downstream oxidation product of guanosine oxidation and detected substantial amounts of those in native and in vivo oxidized RNA. We demonstrate that oxo[8]G is only a transient oxidation product of guanosine oxidation, which does not accumulate but is preferentially consumed in higher-level oxidation reactions.

## Results

### AlkAnilineSeq profiling reveals moderate, but consistent cleavage signals at purine sites in cellular RNAs

After perusing several hundred datasets generated by AAS on RNA across various organisms over the past years, we consistently observed reproducible background signals at purine residues from aniline cleavage, predominantly at guanosine residues. The phenomenon is most easily observed on rRNA because of its high abundance. An example of yeast 25S rRNA is shown in Fig. 1b, where signals are plotted as the so-called stop ratio, a specific measure obtained by the proportion of reads starting at a given position relative to all the reads overlapping that position[56]. Figure 1c shows that about three quarters of the signals above threefold noise level are guanosines, and one quarter are adenosines. Given the almost complete absence of such signals at pyrimidine sites (Fig. 1c), we hypothesized that this purine-specific pattern may stem from the specific chemistry of purine nucleobases rather than reflect random noise. Importantly, the defining characteristic of AAS signals is a chemical alteration of the nucleobase, which weakens its glycosidic bond. This is exemplified by m[7]G, a known modification at position 1575 of yeast 18S rRNA, giving rise to a very strong AAS signal (Supplementary Fig. 1a). To avoid interfering effects from such canonical modifications, which are well documented for rRNA in yeast and other species, the corresponding sites were excluded from further analysis. We hypothesized the origin of the purine signals to be of metabolic nature and thus compared the cellular RNA to an in vitro transcript (IVT) of identical sequence.

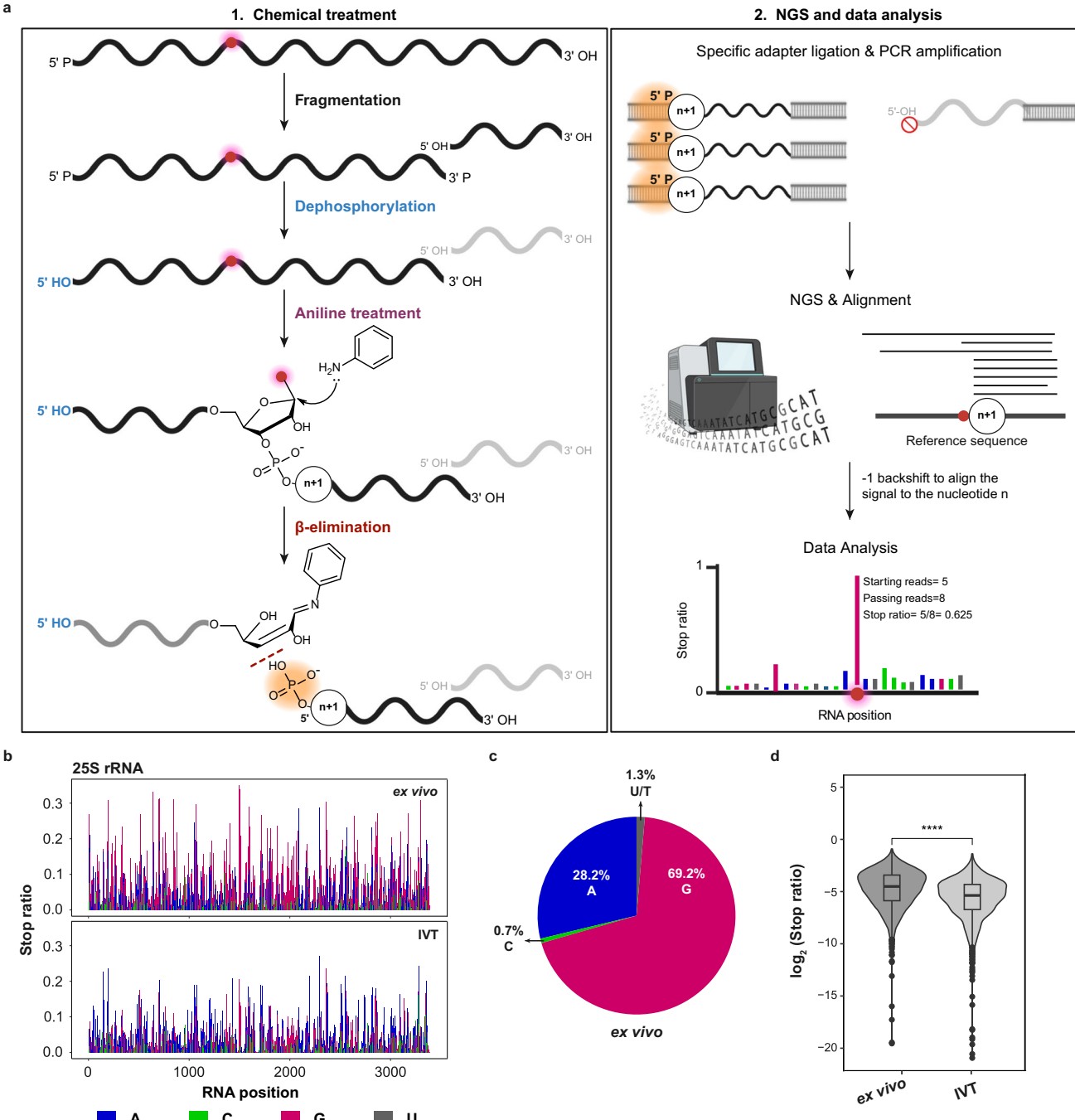

**Fig. 1 | AlkAnilineSeq detects background signals at purine sites in RNA.**
**a** Workflow of the AlkAnilineSeq (AAS) protocol. The left panel (1.) details the initial chemical treatment. This process involves fragmentation of RNA under alkaline conditions using a bicarbonate buffer (pH 9.2) at high temperatures (96 °C). The alkaline treatment also renders it more prone to cleavage of certain modified residues. Following fragmentation, existing 5'- and 3'-phosphates, as well as 2',3'-cyclophosphates in the RNA undergo enzymatic dephosphorylation. Subsequent cleavage with aniline targets the fragile *N*-glycosidic bond, leading to the formation of new RNA fragments with 5'-phosphates. The right panel (2.) visualizes NGS library preparation and data analysis. During library preparation, adapters ligate to the RNA fragments with 5'-phosphates, facilitating the positive selection of these newly generated fragments. These libraries are then PCR-amplified and sequenced. The resulting read fragments are aligned against a reference RNA with a -1 backshift introduced to align the signal to the modified nucleotide. The AAS signals are

indicated by the Stop ratio, derived from the proportion of reads starting at a given position relative to all reads overlapping it. **b** Background AAS profiles of yeast 25S rRNA ex vivo (naked RNA, *n* = 3) compared to IVT yeast 25S rRNA (*n* = 2). RNA positions are indicated on the *x*-axis, and the Stop ratio is depicted on the *y*-axis. **c** Fraction of A, C, G, and U/T for yeast 25S rRNA (ex vivo). **d** The box-violin plot represents AAS signals at G residues across different rRNA species (5.8S, 18S, and 25S) comparing RNA isolated from yeast (ex vivo, left, *n* = 3) and IVT rRNA (IVT, right, *n* = 2). Statistical comparisons were conducted using the Wilcoxon test to evaluate differences across conditions (*p*-value = 2.40 × 10^{-40}). Box plots show the median (centre line), 25th and 75th percentiles (bounds of box), and whiskers indicate the data points within 1.5x interquartile range (IQR). Source data are provided as a Source Data file. Created in BioRender. Raorane, K. (2025) https://BioRender.com/v1o2mt6.

The IVT exhibited markedly reduced signal intensity (Fig. 1b, d), supporting the idea that the observed signals in cellular RNA arose in vivo, i.e., from physiological processes. To probe for the influence of high pH during the alkaline fragmentation step of AAS, we replaced it with $Mg^{2+}$-induced fragmentation. Importantly, the m[7]G1575 signal diminished significantly, showing that detection of m[7]G requires fragmentation at high pH and temperatures (Supplementary Fig. 1a). The fact that background signals at guanosines remained largely unaffected by fragmentation conditions argues against the origin of the signals in metabolic alkylation that would lead to m[7]G[61,62]. We therefore focused on oxidative damage as the most plausible cause of the apparent background. This hypothesis is in line with their redox potentials, guanine being the most easily oxidized nucleobase, followed by adenine. In control experiments, we determined that parameters of the RNA isolation protocol (e.g., hot phenol, Supplementary Fig. 1b) did not influence signal strength, further suggesting that these signals originate from chemical alterations that were inflicted in vivo, rather than during sample preparation.

## LC-MS analysis of major oxidation products shows guanosine as the primary target

To verify if it was indeed oxidative damage that preferentially caused AAS signals at purine residues, we treated RNA with hypochlorous acid (HOCl) and analyzed the resulting RNA oxidation products. Hypochlorous acid was selected specifically for its ability to penetrate cells while being absent from the normal physiology of yeast. This allowed us to use the appearance of chlorinated ribonucleosides in RNA as conclusive proof of oxidation in living cells. Thus, while HOCl is generated in vivo in certain higher eukaryotic cells, this facet of its biology was deliberately shunned here. To identify the products of RNA oxidation in vivo, we first conducted a thorough analysis of in vitro oxidized RNA. The unstable nature of the oxidant made it necessary to employ a rather sophisticated preparation and titration procedure, which requires expressing HOCl concentrations as free $Cl_2$ (Supplementary Fig. 2 and corresponding text in the supplement). In this manuscript, we will use $Cl_2$ as a surrogate to indicate the concentration of active HOCl.

As a standard procedure throughout this study, RNA aliquots were treated with increasing concentrations of HOCl, termed oxidation series. Thereafter, RNA integrity was assessed by denaturing PAGE followed by GelRed staining. Interestingly, RNA bands began to fade at HOCl concentrations corresponding to ~50 μM $Cl_2$ and completely vanished at concentrations above 100 μM $Cl_2$, as exemplified with a synthetic 38mer (Fig. 2a), IVT of tRNA[Asp] and yeast total RNA (Supplementary Fig. 3a, b). We could show via [32]P-labeling that the RNA chain itself remained intact even when bands were no longer visible by staining (Fig. 2b). This failure to be stained by an intercalating reagent, previously described for DNA[63], correlated with a decrease of UV-chromophores (Fig. 2c), suggesting a decrease in RNA base content. Further LC-MS analysis confirmed the decrease in the content of purine bases upon HOCl treatment. Figure 2d shows the structures of major oxidation products identified after HOCl treatment of mononucleosides, which were consistent across in vitro oxidation reactions of synthetic oligonucleotides, in vitro transcripts (IVTs), and finally in RNA isolated from HOCl-treated living yeast. Figure 2e, f show LC-MS results of a typical RNA oxidation series, here exemplified with a synthetic model oligoribonucleotide. As oxidant concentrations rose, the content of nucleobases declined (Fig. 2e), accompanied by an increase in simple purine oxidation products (Fig. 2f). The LC-MS data revealed differential reactivities among the four single nucleosides consistent with their redox potentials. Correspondingly, uridine reacted slowly, yielding minor amounts of Cl[5]U, while cytidine gave equally low amounts of ho[5]C but nearly quantitative amounts of Cl[5]C. Both purines yielded 8-oxo and 8-chloro derivatives, with guanosine showing higher reactivity, aligning with its predominance in AAS signals (Fig. 1c,

Supplementary Fig. 4a–c). These observations were consistent across nucleosides (Supplementary Fig. 4b, c), oligonucleotides (Fig. 2e, f), IVTs (Supplementary Fig. 4d, e) and native RNAs (Supplementary Fig. 4f, g). Of significance to the loss of staining and the changes of UV absorption (Fig. 2a, c), we identified several advanced guanosine oxidation products, including rDiz, rZ, rIz, r2Ih, rGh, and rSp (summarized in Supplementary Table 1 and depicted in Supplementary Fig. 5a). These are categorized as ring-opened guanosine derivatives for the purpose of this study. Minor variations in oxidation series were noted between different RNAs, likely reflecting differences in nucleobase composition (vide infra) and RNA structure. Based on these results, we selected 50 μM free $Cl_2$ as the relevant concentration for subsequent investigations and as reference for further investigations.

Comparative LC-MS analyses of RNA isolated from yeast cells after in vivo oxidation with HOCl revealed highly similar characteristics, including the presence of chlorinated nucleosides that clearly demonstrate exogenous HOCl-induced RNA oxidation inside cells. Figure 2g shows the content of major nucleosides, Cl[5]U and Cl[5]C following in vivo oxidation with HOCl at elevated concentrations (0–20 mM $Cl_2$) compared to concentrations used in vitro (0–500 μM $Cl_2$). These higher HOCl concentrations were required to overcome the quenching effects of reducing agents in the growth media and the yeast cytosol. Although purine destruction complicated normalization, the data recapitulates the eventual conversion of all cytidines to Cl[5]C. Figure 2f, h and Supplementary Fig. 4e, g further demonstrate the key observation of successive destruction of purine rings with peak levels of 8-oxo and 8-chloro purines at 25-50 μM $Cl_2$ in vitro, and 1 mM $Cl_2$ in vivo, respectively. As these oxidation products decline at higher oxidant concentrations, the abundance of ring-opened guanosine derivatives increases accordingly, in line with them being higher oxidation products of oxo[8]G (Supplementary Fig. 5b, c).

## Oxidation preferentially proceeds at guanosines via oxo[8]G and leads to abasic sites

We subsequently picked up on the earlier observation that nucleoside composition influences the observed RNA oxidation profiles. We performed oxidation series of synthetic oligonucleotides of binary or ternary composition, i.e. containing either two or three of the four major nucleosides (see Supplementary Table 5). The interpretation is most straightforward for those 3 binary permutations (out of 6) that contain uridines, as the near inertness of this nucleoside allows an assessment of the partner nucleoside alone. As shown in Fig. 3a–c, the content of all cytidines, adenosines, and guanosines decreased similarly within an oxidation series, indicating that uridines are relatively inert and that oxidation predominantly targets the non-uridine nucleobase. Concomitantly, the content of oxidized and chlorinated products increased (Fig. 3d–f). Interestingly, when examining a binary oligonucleotide composed only of adenosines and guanosines (Fig. 3g, h), the presence of guanosine appeared to protect adenosines from oxidation, since a decrease of A content was observed only at higher $Cl_2$ concentration (Fig. 3b, g). However, this protective effect was not consistently observed in binary oligonucleotides containing cytidine in combination with either adenosine or guanosine (Supplementary Fig. 6a, b). The AC binary oligonucleotide was relatively stable, while the GC binary RNA was substantially damaged at 50 μM $Cl_2$. This discrepancy likely reflects the more similar redox potentials of cytidine and purines. Further analysis of oligonucleotides containing single guanosine and adenosine residues separated by uridine stretches of varying length revealed that adenosine oxidation occurred primarily after guanosine was fully oxidized, except when the purines were adjacent to one another (Supplementary Fig. 6c–f). While a comprehensive understanding requires further investigation, these results demonstrate that both nucleoside composition and sequence context shape oxidation profiles, with guanosine oxidation playing a crucial role.

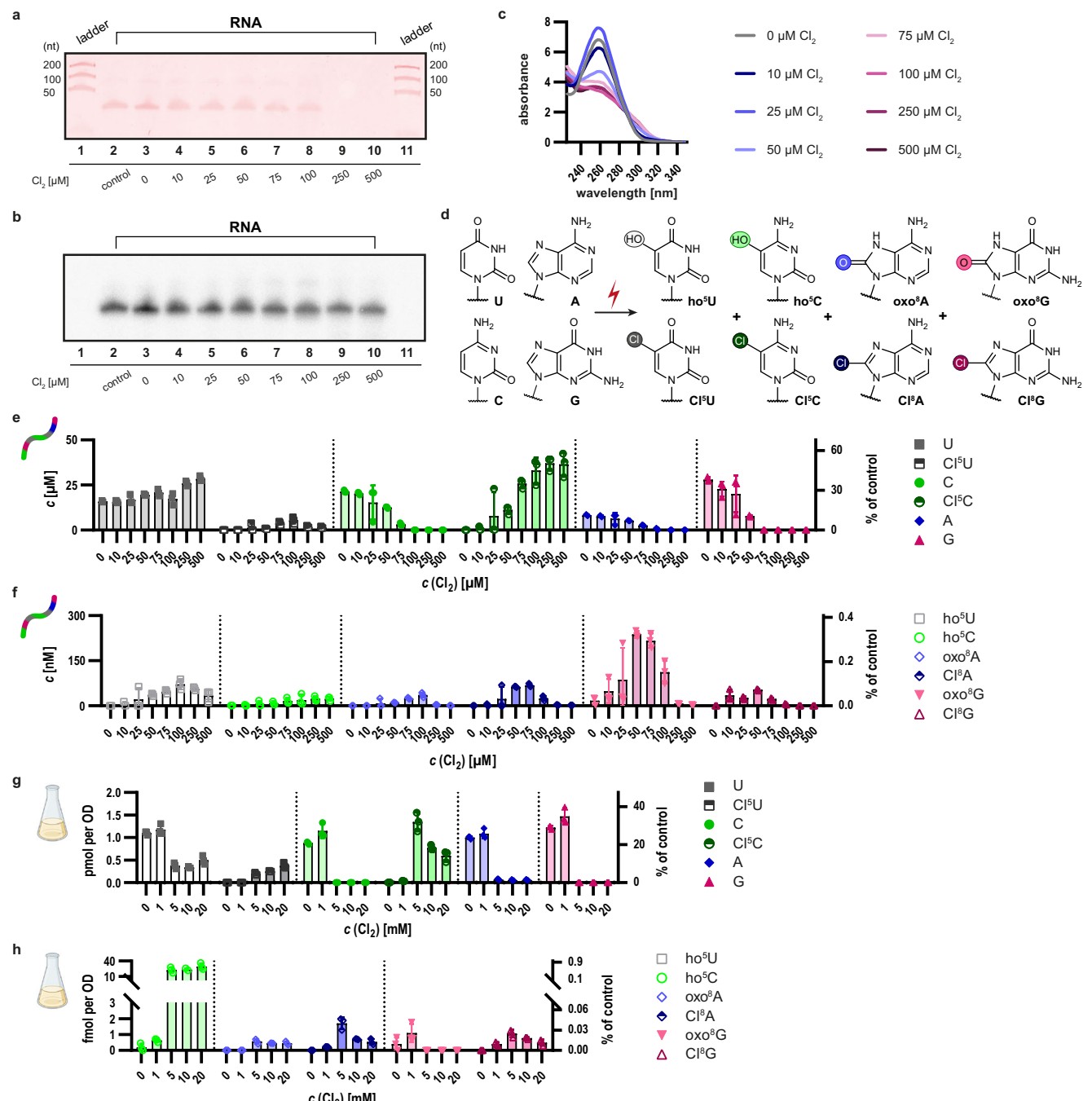

**Fig. 2 | HOCl induces concentration-dependent RNA damage and formation of distinct nucleoside oxidation products in vitro and in vivo. a, b** Denaturing PAGE of a $^{32}$P-labeled synthetic oligonucleotide treated in an in vitro HOCl oxidation series at concentrations ranging from 0 to 500 μM Cl$_2$. Lanes 1 and 11 contain unlabeled ladders indicating 50, 100, and 200 nucleotides (nt). Lane 2 contains an untreated control, while lanes 3 to 10 contain samples oxidized from 0 to 500 μM Cl$_2$. 100 ng was loaded for each sample. RNA was visualized using GelRed™ staining (**a**) and a $^{32}$P-phosphor scan (**b**). **c** Corresponding absorbance spectra of oxidized (0 - 500 μM Cl$_2$) RNA samples after purification measured on a Nanodrop™ spectrophotometer. **d** Structures of HOCl oxidation products. Main nucleosides: uridine (U in gray), cytidine (C in green), adenosine (A in blue), guanosine (G in pink) and their corresponding oxidation products: 5-hydroxyuridine (ho$^5$U in light gray), 5-chlorouridine (Cl$^5$U in dark gray), 5-hydroxycytidine (ho$^5$C in light green), 5-chlorocytidine (Cl$^5$C in dark green), 8-oxoadenosine (oxo$^8$A in light blue), 8-chloroadenosine (Cl$^8$A in dark blue), 8-oxoguanosine (oxo$^8$G in light pink) and 8-chloroguanosine (Cl$^8$G in dark pink). Chemical structures are drawn with ChemDraw. **e** Absolute quantification of nucleosides derived from the oxidized oligonucleotide via LC-MS/MS using external

standards for calibration. The synthetic, unmodified RNA oligonucleotides were subjected to HOCl treatment at the indicated concentrations. Nucleoside concentrations are indicated in μM relative to the volume of the oxidation reaction. The right y-axis is normalized to the total nucleoside content of the untreated control. Note that the apparent increase in pyrimidines at high concentrations of Cl$_2$ results from a UV normalization bias caused by purine chromophore oxidation. **f** Low-abundant oxidation products are presented in nM concentrations. **g** Absolute quantification of a HOCl concentration series performed in vivo on *S. cerevisiae* cells. Cl$_2$ concentrations are indicated on the x-axis and range from 0 to 20 mM Cl$_2$. Total RNA was extracted and subjected to nucleoside analysis. The left y-axis indicates nucleoside concentrations in pmol per OD, and the right y-axis shows relative proportions normalized to the total nucleoside concentration in the untreated control. **h** Absolute amounts of the oxidized nucleosides displayed in fmol per OD on the left y-axis. The right y-axis is normalized to the total nucleoside concentration in the untreated control. Data are presented as the mean of three biological replicates ± SD, n = 3. Source data are provided as a Source Data file. **g, h** Created in BioRender. Weber, M. (2025) https://BioRender.com/ap8gmkc.

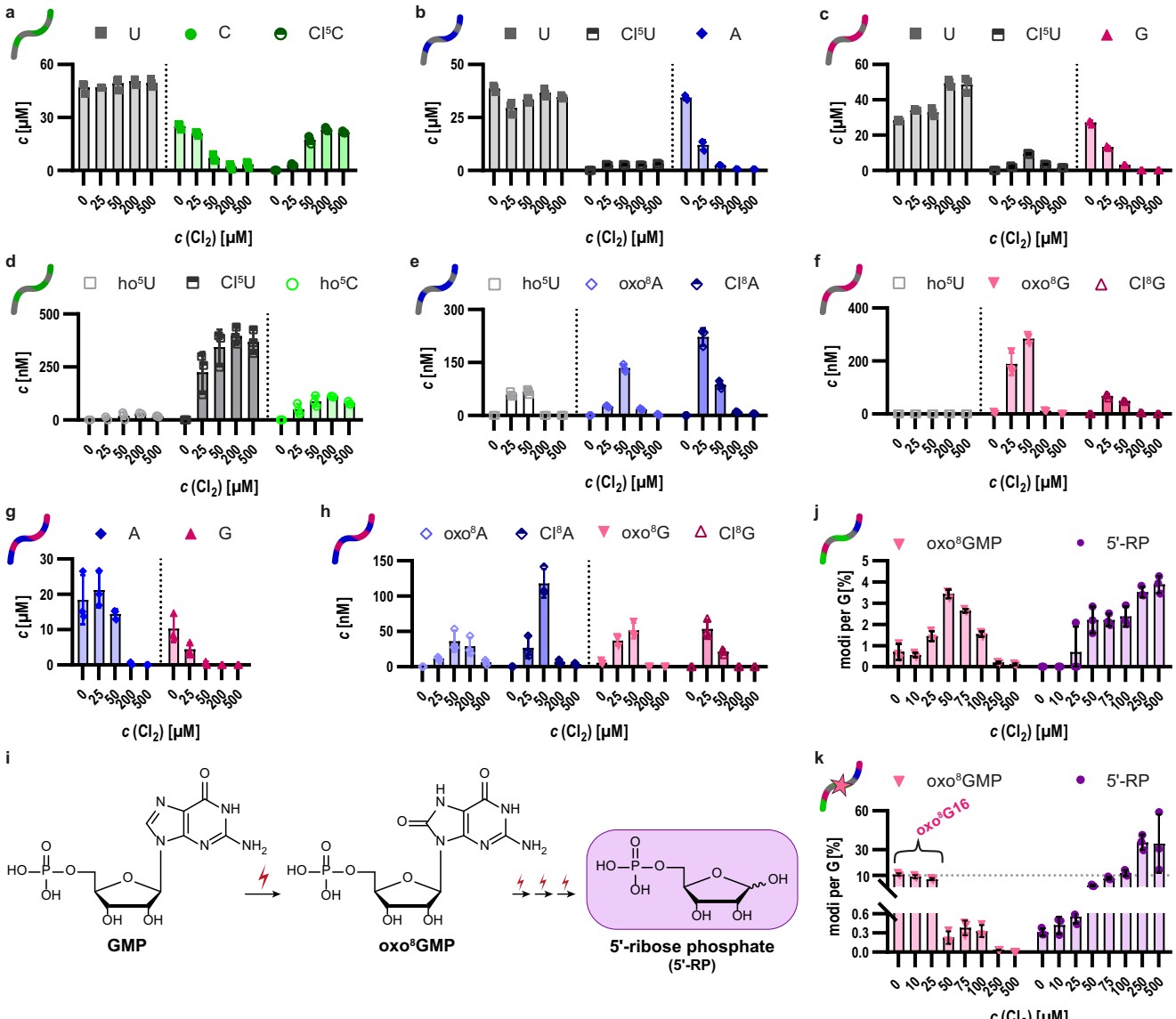

**Fig. 3 | Absolute quantification of oxidative lesions and abasic sites in HOCl-treated synthetic oligoribonucleotides. a–c** Binary oligonucleotides composed of U and either C, A or G were subjected to HOCl oxidation (0 to 500 μM Cl₂). Quantification of parent [U (gray), C (green), A (blue), G (pink)] and oxidized nucleosides [Cl⁵C (dark green), Cl⁵U (dark gray), ho⁵C (light green), ho⁵U (light gray), Cl⁸A (dark blue), oxo⁸A (light blue), Cl⁸G, (dark pink), oxo⁸G (light pink)] was performed using LC-MS/MS and external standards for calibration purposes. Absolute amounts are indicated in μM concentrations on the left y-axis relative to the initial reaction volume. The results of the CU combination are depicted in **a**, the AU combination in (**b**) and the GU combination in (**c**). **d–f** Absolute amounts of oxidation products presented in nM concentrations of the CU combination (**d**), the AU combination (**e**) and the GU combination (**f**). **g, h** LC-MS/MS quantification of a binary AG composition expressed in μM for both parent nucleosides (**g**) and in nM for the corresponding oxidation products (**h**). **i** Proposed progression of multiple oxidation events preferentially occurring at guanosines, ultimately yielding abasic sites denoted as 5′-ribose phosphates (5′-RP). Chemical structures are drawn with ChemDraw. **j** Absolute quantification of 5′-RP (light purple) and oxo⁸GMP (light pink) of an unmodified 38mer oxidized in the range of 0 to 500 μM Cl₂ using LC-MS/MS and external calibration. The amount is normalized to the injected amount of RNA and the number of guanosines present in the sequence. The 5′-RP shows steadily increasing amounts whereas oxo⁸GMP is present at maximum concentrations at 50 μM Cl₂. **k** Absolute quantification of 5′-RP (light purple) and oxo⁸GMP (light pink) of an oxo⁸G-modified 38mer at position 16 oxidized in the range of 0 to 500 μM Cl₂ using LC-MS/MS and external calibration as in **j**. The amount is normalized to the injected amount of RNA and the number of guanosines present in the sequence. The decreasing oxo⁸GMP signal at 0, 10, and 25 μM Cl₂ originates from the oxo⁸G site at position 16, whereas oxo⁸GMP signals at higher applied oxidant concentrations presumably correspond to guanosines other than G16. LC-MS data of n = 3 biological replicates are shown as mean values ± SD. Source data are provided as a Source Data file.

Given the prevalence of guanosine-derived signals in sequencing and the preferential oxidation of guanosines evidenced by LC-MS, we revisited the role of oxo⁸G. In various oxidation series (Fig. 2e–h, Fig. 3), oxo⁸G appeared as guanosine oxidation progressed, but diminished at higher oxidant concentrations. This pattern suggested that oxo⁸G is not a stable end-product, but rather an intermediate that undergoes further chemical reactions. This interpretation was reinforced by experiments using a synthetic oligonucleotide containing a single oxo⁸G at position 16 (Supplementary Fig. 6g, h). Here, the level of oxo⁸G (Supplementary Fig. 6h) is high at the onset but decreased markedly at 25 μM oxidant before rising again at 50 μM and disappeared at higher oxidant concentrations. Together, these findings demonstrate that oxo⁸G is rapidly formed but also rapidly consumed, supporting the hypothesis of its role as a key intermediate during the formation of other oxidation products such as ring-opened guanosine derivatives.

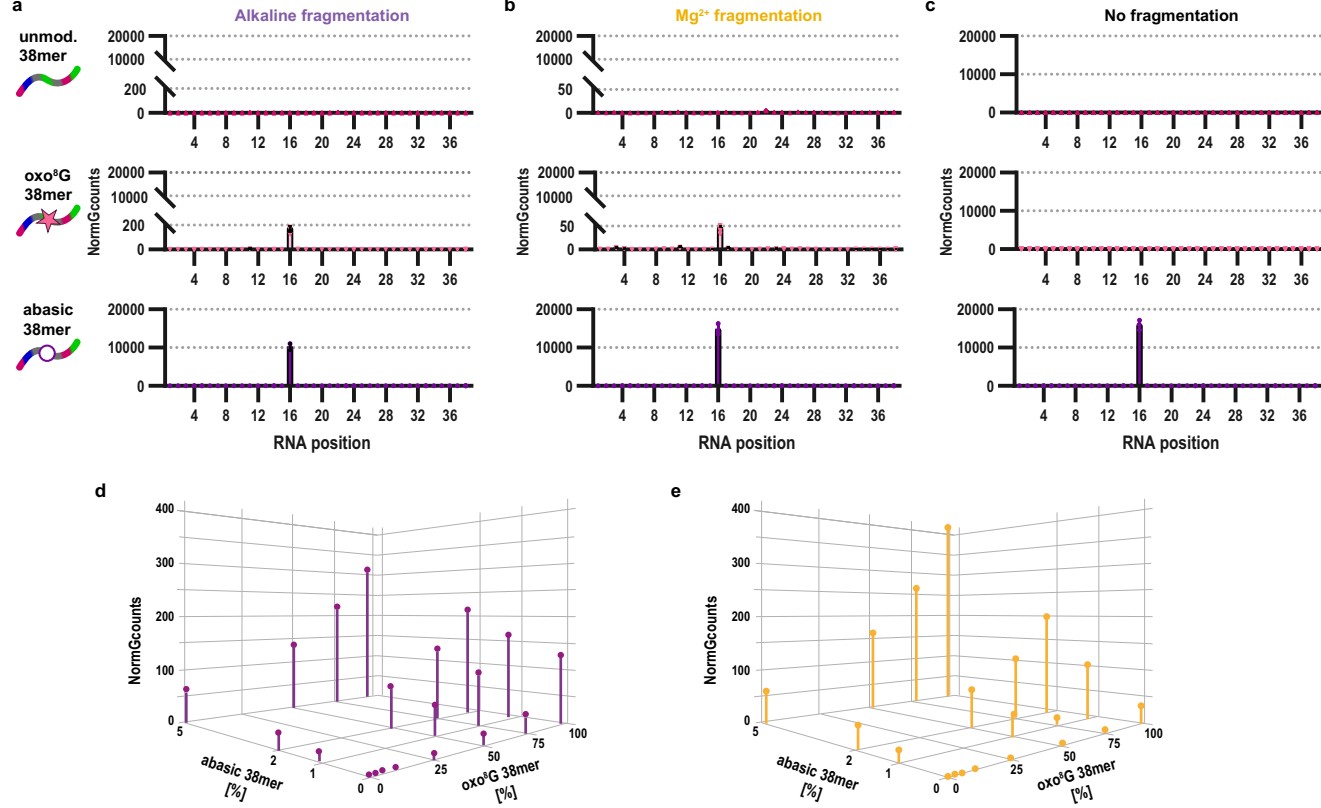

**Fig. 4 | Abasic Sites, not oxo⁸G, dominate aniline cleavage–dependent RNA oxidation signals.** **a–c** Comparative RNA sequencing profiles of an unmodified RNA 38mer (top panel), its oxo⁸G-congener (middle panel) and abasic-congener (bottom panel) under distinct fragmentation conditions including alkaline (**a**) and Mg²⁺ (**b**) treatment in addition to no fragmentation (**c**), respectively. The $y$-axis indicates NormGcounts and the $x$-axis indicates RNA positions. Data of biological replicates are presented as mean ± SD, $n = 3$ for unmodified and abasic 38mers and $n = 6$ for the oxo⁸G-modified 38mer. **d, e** Signal strength of 38mers mixed in different ratios analyzed by AAS (**d, left panel**) and OAbSeq (**e, right panel**) methods.

The three axes indicate the proportion [%] of the oxo⁸G 38mer, the abasic 38mer and the respective signal strength in NormGcounts. The unmodified 38mer was added to complete the mixture to 100%. Due to the 5′ specific ligation enrichment step during the library preparation protocol (see Methods section), AAS scores, including NormGcounts, do not show a linear dependence on the RNA modification level. Thus, the signals obtained for mixtures of oligonucleotides may not be strictly additive. Data points of one replicate are shown. Source data are provided as a Source Data file.

Based on these findings, it seemed plausible that the dominant guanosine-derived signals in AAS might originate from progressive guanosine oxidation beyond oxo⁸G, potentially all the way to abasic sites (Fig. 3i). Indeed, oxidized RNA reacted with an aldehyde-reactive probe (ARP), a dye-conjugated hydroxylamine commonly used to detect abasic sites in DNA[55,64,65]. As shown in Supplementary Fig. 7, oxidized RNA displayed notable ARP reactivity consistent with the presence of aldehyde groups at abasic sites. To directly confirm abasic site formation, LC-MS analysis of the oxidized model oligoribonucleotide was performed (Fig. 2e, f), but after enzymatic digestion to monophosphates rather than to nucleosides. Quantification of abasic sites in the form of 5′-ribose phosphates (5′-RP) and oxo⁸GMP on nucleotide level is shown in Fig. 3j. Guanosine levels declined continuously with increasing concentrations of HOCl (Fig. 2e), while oxo⁸G transiently appeared before disappearing (Figs. 2f, 3j). In parallel, levels of 5′-RP steadily increased indicating continuous formation of abasic sites as a downstream product of guanosine oxidation (Fig. 3j). A similar pattern was observed for a synthetic 38mer carrying a single oxo⁸G at position 16 (Fig. 3k). Upon addition of oxidant, the oxo⁸G residue was rapidly consumed followed by a slight increase attributed to oxidation of guanosines at positions other than 16. Importantly, the final content of abasic sites in this oxo⁸G16-containing oligonucleotide even exceeded that of its unmodified counterpart, providing evidence that both oxo⁸G and canonical G residues were converted into abasic sites under oxidative conditions. Additionally, considering the results shown in Fig. 2, these findings support the mechanistic pathway shown in Fig. 2, these findings support the mechanistic pathway

proposed in Fig. 3i: Guanosine is oxidized to oxo⁸G before undergoing further oxidation to eventually yield abasic sites. Although the exact identity of eventual intermediates, e.g., from Supplementary Fig. 5a remains to be clarified, it is clear that abasic sites represent the predominant and stable product of RNA oxidation exceeding the content of oxo⁸G (Fig. 3j, k).

The data presented so far supports that the majority of abasic sites in oxidized RNA derive from guanosines rather than adenosines. In agreement, the guanosine content of RNA drops faster than that of adenosine (*vide supra*). For example, at the reference conditions of 50 μM Cl₂ in vitro and 1 mM Cl₂ in vivo, a very small fraction of adenosines was yet oxidized, while oxidation of guanosine was much more evident. Consequently, oxo⁸A was only detected in low amounts, which correlated with the lower number of AAS signals mapped to adenosines (Fig. 1c). Moreover, analysis of AAS profiles using other proposed scores demonstrated that the most intense signals are >95% at the guanosine residues. We have therefore focused on guanosine oxidation in what follows.

### Reactivity of RNA in synthetic model RNA oligonucleotides containing oxo⁸G and rAP sites

Having established a plausible molecular basis for the preferential occurrence of AAS signals at guanosine sites, we asked for the chemical species actually giving rise to the observed signals (Fig. 1). To this end, we applied the AAS protocol to the above model 38mer RNA and congeners of the same sequence: an unmodified sequence, a variant

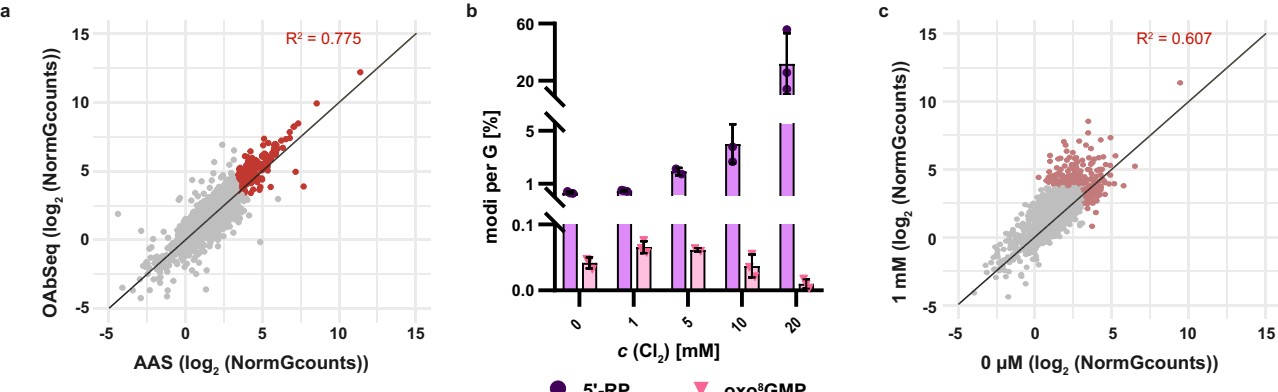

**Fig. 5 | In vivo oxidation already occurs under physiological conditions as analyzed via AAS, OAbSeq and LC-MS. a** Correlation plot comparing AAS and OAbSeq scores of yeast RNA oxidized in vivo with 1 mM $Cl_2$. Sites that are three times above noise level are colored red. Gray points represent sites where either AAS, OAbSeq, or both scores fall below the specified threshold (NormGcounts = 10). Data points of three biological replicates ($n = 3$), including 1490 G residues, are shown. **b** Absolute quantification of 5′-RP (purple) and oxo[8]GMP (light pink) of *S. c.*

total RNA using external calibration and LC-MS/MS. Absolute amounts are normalized to the injected amount during LC-MS and the proportion of G in total RNA as described in the Methods section. Shown are mean values ($n = 3$) ± SD of biological triplicates. **c** Correlation plot of AAS signals in untreated (0 μM $Cl_2$) and HOCl treated (1 mM $Cl_2$) RNA. Sites that are three times above the noise level are colored red. Data points of three biological replicates ($n=3$), including 1490G residues, are shown. Source data are provided as a Source Data file.

with a single oxo[8]G at residue 16 and another variant carrying an abasic site at the same position. The AAS cleavage signal reflects the number of aligned reads starting at a given position of a given RNA. This score accounts for the differential reactivity of specific nucleotides within the RNA sequence and can be normalized in different ways. Since prominent signals were obtained almost exclusively at guanosine sites, we developed a specific score termed Normalized G Counts, adapted for AAS data[66]. This score represents the AAS cleavage signals normalized to the median cleavage signal of surrounding non-G residues in an 11-nucleotide window. As shown in Fig. 4a, the unmodified 38mer caused no detectable AAS signal, while the oxo[8]G congener showed a clear signal (~200 NormGcount units), and a very strong signal was detected for the abasic 38mer (~10,000 NormGcount units). The latter had been anticipated as a result of efficient cleavage via $\beta$-elimination under aniline treatment, while the less efficient AAS cleavage at oxo[8]G was in keeping with previous reports[51]. However, for both cases the effect of the alkaline treatment needed clarification. Hence, this step was replaced by high-temperature RNA fragmentation with magnesium ions. This alternative treatment also fragments long RNA to a size compatible with library preparation for Small RNAseq, but avoids high concentrations of hydroxide anions, which may displace the nucleobase from the ribose, resulting in abasic sites. The corresponding profile for the $Mg^{2+}$-fragmented oxo[8]G-38mer in Fig. 4b shows a clear ~4-fold signal depression from 200 to 50 NormGcount units, while the signal for the abasic site did not diminish. The control experiment, which omitted any fragmentation step, included only aniline cleavage (Fig. 4c). This proved successful cleavage exclusively for the abasic oligomer, in line with guanosine and oxo[8]G not reacting with aniline. To delineate the contribution of either lesion to the AAS signal, we analyzed defined mixtures of the three RNA oligonucleotides in varying stoichiometries. As shown in Fig. 4d, the oxo[8]G contribution to the overall AAS signal strength became discernible when its proportion exceeded 20% stoichiometry, which is vastly more than was measured by LC-MS in any of the investigated RNAs. Thus, abasic sites contribute much stronger to the overall AAS signal than oxo[8]G, even when present in much lower molar ratios and the fraction originating from oxo[8]G can be essentially neglected. Indeed, given that abasic sites are more frequent than oxo[8]G (*vide supra*), we conclude that the majority of AAS signals observed for in vitro or in vivo oxidized RNA must originate from abasic sites rather than from oxo[8]G. This applies even more to the protocol involving $Mg^{2+}$ treatment instead of standard alkaline hydrolysis (Fig. 4e). We termed this modified protocol OxiAbasic

Sequencing (OAbSeq), because among the RNA oxidation products, it almost exclusively reveals abasic sites.

## In vivo HOCl oxidation preferentially hits sites already oxidized under physiological conditions

To understand the biological relevance of our findings, we performed site-specific mapping of HOCl-inflicted guanosine lesions in rRNA using AAS and OAbSeq. These in vivo data derive from the same *S. cerevisiae* samples previously characterized by LC-MS (Fig. 2g, h), comparing mock-treated *versus* cells to those exposed to 1 mM $Cl_2$. Figure 5a shows a correlation plot of signals from AAS and OAbSeq of the in vivo oxidized samples. The high correlation ($R^2 = 0.775$) indicates that oxo[8]G contributes minimally to the overall picture. Otherwise, AAS would generate stronger signals than OAbSeq (compare Fig. 4a, b). This interpretation was further supported by LC-MS data of a nucleotide digest (Fig. 5b), which showed abasic sites to be at least an order of magnitude more abundant than oxo[8]GMP in the untreated yeast, and the predominance of abasic sites significantly increased upon in vivo oxidation with $Cl_2$. Next, we investigated if the signals mapped to oxidized guanosines after oxidation at 1 mM $Cl_2$ in vivo corresponded to those already oxidized in vivo to a low degree without treatment (i.e, those shown in Fig. 1). Figure 5c shows a correlation plot between the naive in vivo sample (0 μM) and the one oxidized in vivo using 1 mM free $Cl_2$. The shape of the points' cloud reflects stronger oxidation of the latter with a substantial number of data points above the diagonal indicating increased signals after HOCl exposure. Importantly, this plot demonstrates that HOCl oxidation preferentially targets the same guanosine residues that are already susceptible to endogenous oxidation. In turn, the data supports the conclusion that these sites are naturally prone to oxidation, and likely subject to ongoing physiological oxidation by oxidants such as ROS.

## Comparative analysis of rRNA oxidized in vivo and in vitro

We further extended our HOCl mapping analysis from in vivo to in vitro oxidized samples of naked rRNA and to in vitro oxidized isolated ribosomes. To obtain comparable signal patterns for rRNA in isolated ribosomes, we adjusted HOCl concentrations accordingly. Once more, we found little difference between AAS and OAbSeq in correlation plots (Supplementary Fig. 8a, b). The violin plots in Fig. 6a show clear differences in signal distribution of oxidized and non-oxidized samples in all three contexts. Analysis of signals in naked rRNA indicated some preference for oxidation of guanosines in single-

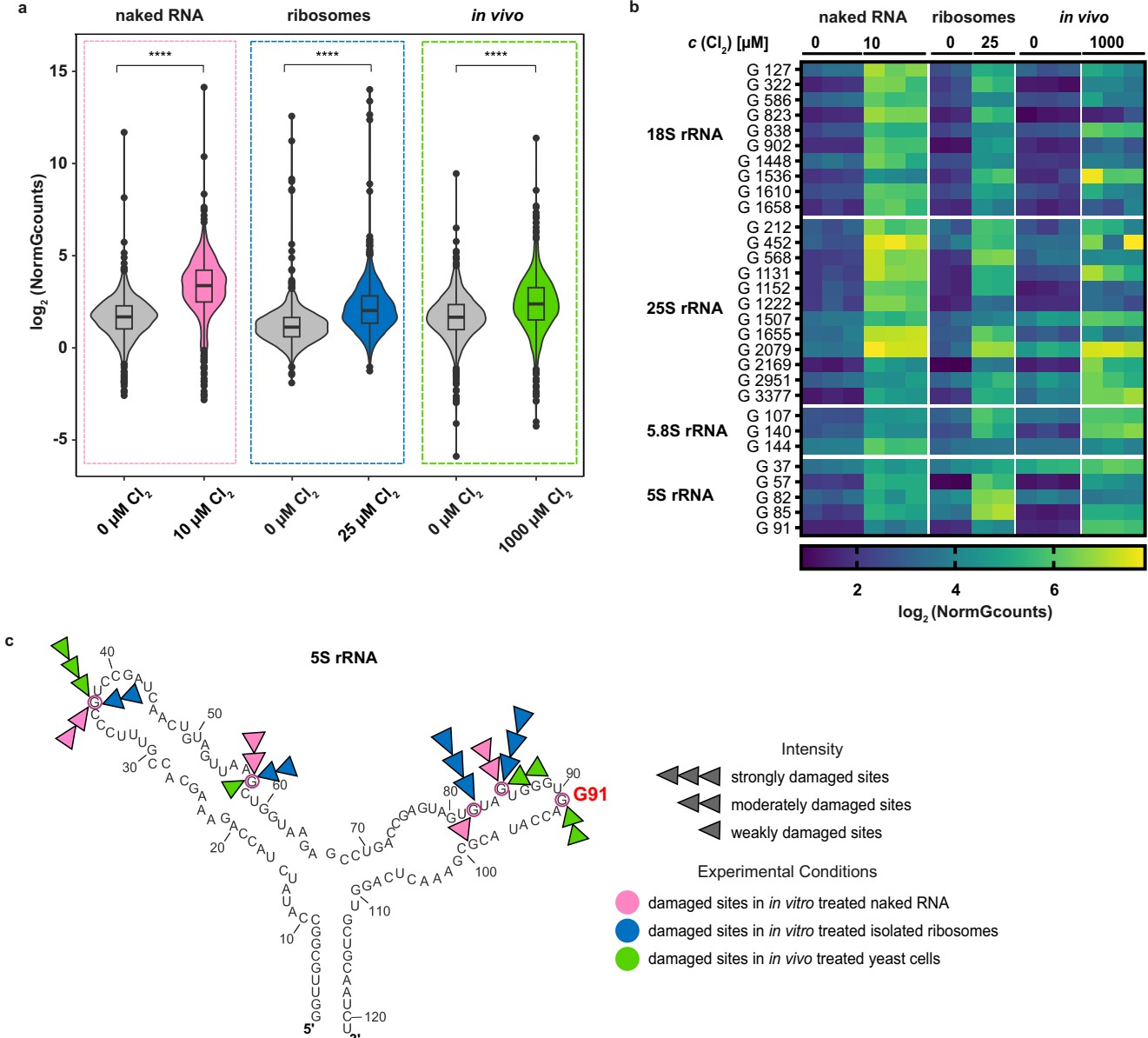

**Fig. 6 | Differential susceptibility of rRNA to HOCl-mediated oxidation in naked RNA, ribosomes and living yeast cells. a** The box-violin plot compares global RNA oxidation patterns in yeast RNA oxidized with HOCl under the different experimental conditions: in vitro isolated rRNA (naked RNA, $n = 3$, $p$-value $= 5.36 * 10^{-184}$, biological replicates) depicted in pink, in vitro oxidation of ribosomes (RNPs, $n = 2$, $p$-value $= 7.76 * 10^{-98}$, biological replicates) depicted in blue and in vivo oxidation of rRNA isolated from living yeast cells ($n = 3$, $p$-value $= 9.81 * 10^{-53}$, biological replicates), depicted in green. The oxidized RNA samples were subsequently analyzed using AAS, and the data are represented by average NormGcounts. Statistical comparisons were conducted using the Wilcoxon test to evaluate differences between treated and untreated samples, including 1436 data points per condition. Box-violin plots show the distribution of $\log_2$ (NormGcounts). Box plots inside violins indicate the median (centre line) and the 25th – 75th percentiles (box), and whiskers indicate the data points within 1.5x interquartile range (IQR). **b** The heatmap indicates guanosine oxidation patterns for 30 most severely damaged sites, which were identified following AAS in response to the aforementioned experimental settings. The color key for $\log_2$ (NormGcount) is provided in the insert below. Each column represents an individual biological replicate, and each row represents the nucleotide position on the rRNA that is differentially damaged. Of note, 18S rRNA contains the natural, stoichiometrically modified m$^7$G1575 residue, which gives a robust AAS signal; thus, this position was ignored in the analysis. **c** A detailed insight into the spatial distribution of oxidation signals across the secondary structure of 5S rRNA. The strength of oxidation signals in response to diverse experimental settings is color-coded and mapped differently. The secondary structure was retrieved from RiboVision2[78] for mapping. Source data are provided as a Source Data file.

stranded regions, albeit with imperfect correlation (Supplementary Fig. 8c). This trend suggests some protection of guanosines by structural features, akin to structural probing with alkylating and acylating agents[8,67]. A comparative heat map of guanosine oxidation signals from the three oxidation conditions is displayed in Fig. 6b. Not unexpectedly, certain G residues appear less oxidized in ribosome RNPs and in living cells, compared to naked RNA when exposed to HOCl.

However, there are several exceptions arguing against a general rule such as rRNA binding to proteins in the ribosome would generally protect RNA from oxidation. This is illustrated by a few selected examples: a first case in point consists of G82 and G85 in 5S rRNA (Fig. 6c), which are most reactive in isolated ribosomes, although one might expect similar protein contacts also in living cells. In another case, G91 in 5S rRNA, highlighted in Fig. 6c is markedly more oxidized

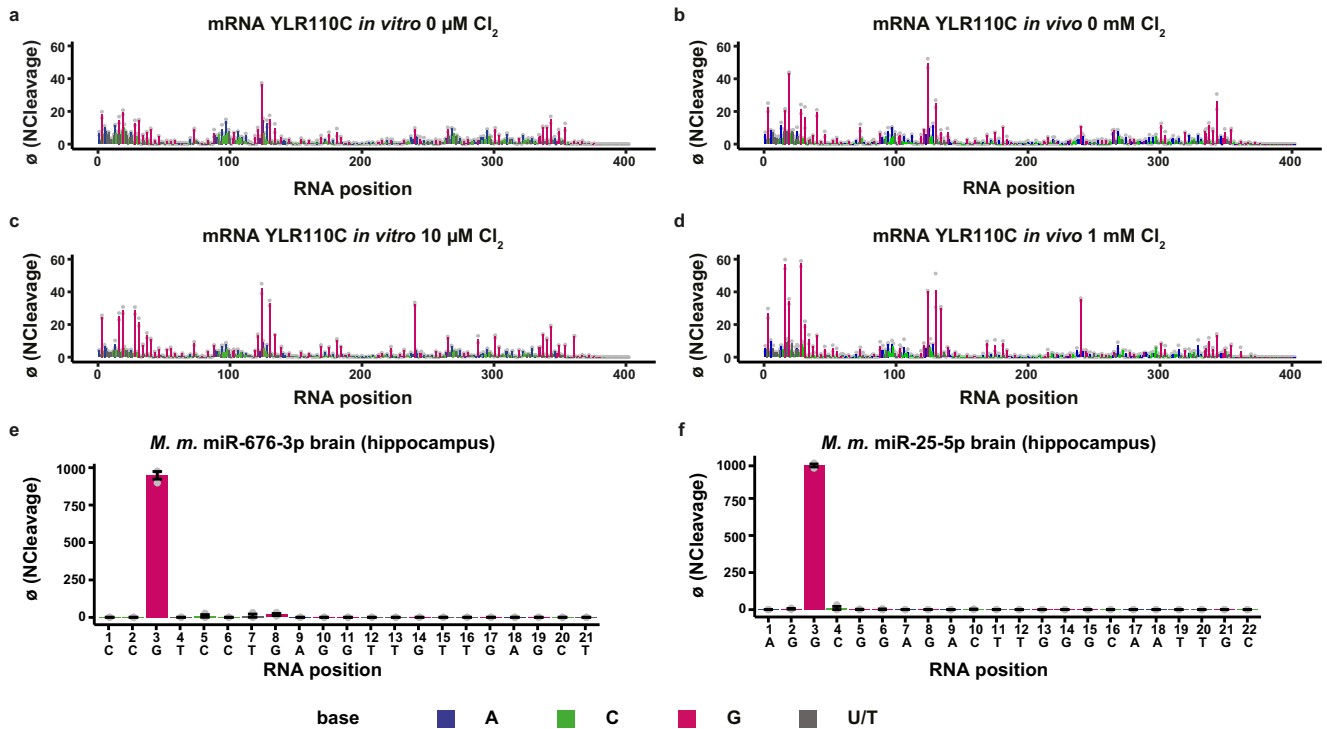

**Fig. 7 | AAS detects preferentially G sites in native mRNA and miRNA under physiological conditions and under oxidative stress. a–d** AAS sequencing profiles of the *S.c.* YLR110C mRNA are plotted as NCleavage (*y*-axis) per position (*x*-axis) under control and oxidative conditions. The respective profiles of control samples for in vitro (**a**) and in vivo (**b**) are shown, as are the oxidized sample profiles for 10 μM Cl$_2$ in vitro (**c**) and 1 mM Cl$_2$ in vivo (**d**). Guanosine-specific signals are prevalent in all conditions and increase with HOCl oxidation. **e, f** Mouse miRNA sequencing profiles reveal strong signals at G3 in the seed region for miR-676-3p (**e**) and miR-25-5p (**f**) from brain tissue (hippocampus). Shown are mean values of Normalized cleavage (NCleavage = reads starting at a given position x 1000/total number of reads aligned to this RNA), which are used for AAS-score-based profiles of biological replicates ± SD (*n* = 2 for *S. c.* mRNA and *n* = 3 for *M. m.*). Source data are provided as a Source Data file.

in living cells, than in naked rRNA and for rRNA in isolated ribosomes. Thus, any ideas about the underlying molecular basis for increased or reduced oxidation at a given RNA site are speculation at this point. However, the aggregated data clearly point out that our method reveals unexpected differences among the three environments, which bear further investigations. Of note, the first dimension of a three-way PCA shows the naked rRNA, distant from the two other environments, while indicating similarities of rRNA bound in ribosomes in vitro and in vivo (Supplementary Fig. 9). We anticipate that the reproducible differences between the two latter environments will make an interesting topic for a fascinating follow-up study.

### Extension to other RNA species

Having characterized and validated AAS and OAbSeq on abundant rRNA, the scope of investigations was expanded to other RNA species. Initial analysis of existing datasets revealed sufficient sequencing coverage to characterize SCR1 (a noncoding RNA found in the signal recognition particle SRP[68]) in untreated (Supplementary Fig. 10a) and in vitro-oxidized (Supplementary Fig. 10b, c) yeast total RNA samples. Treatment with HOCl generates signals at G residues in RNA (one U was also found to be reactive), similar to those found in rRNA. Consistent with rRNA findings, AAS and OAbSeq profiles for SCR1 show substantial similarity, but some more intense signals obtained with the AAS protocol might indicate an increased presence of $oxo^8G$ in addition to rAP sites in SCR1 RNA. Not surprisingly, sequence coverage of mRNA sequences in total RNA preparations was insufficient in the existing datasets. However, the extraction of polyadenylated RNA from remaining aliquots of the experiments shown in Fig. 5c and subsequent AAS and OAbSeq yielded enough reads to characterize some abundant mRNAs, of which YLR110C is displayed in Fig. 7a-d. A comparison of existing guanosine signals of untreated in vitro (Fig. 7a)

and in vivo (Fig. 7b) samples with oxidized in vitro (Fig. 7c) and in vivo (Fig. 7d) samples showed substantial enhancement of preexisting signals upon oxidation, again recapitulating the observations made with rRNA. One guanosine at position 240 responded strongly to HOCl oxidation in vitro and in vivo, while only showing a low signal in the controls.

Finally, extending our sequencing method to animals, we broadened our analysis to some of the most abundant miRNAs from mice. Figure 7e, f display AAS profiles for two miRNAs with a strong guanosine signal in the seed region. These findings are consistent with reports on miRNA oxidation[9,10] and extend the scope to include in vivo data from animals, as the RNA samples were extracted from the mouse brain. However, at this point, the analysis is restricted to the 5'-half of miRNAs, since signals nearer to the 3'-end would correspond to shorter reads of <8 nt, which might not be amenable to unambiguous mapping. Future investigations into optimizing sample preparation may help to overcome this limitation. Nevertheless, the present study proves that AAS and/or OAbSeq can be applied to stable and unstable RNAs from in vivo samples, enabling physiological investigations into oxidatively damaged RNA.

## Discussion

Current literature of oxidatively damaged RNA mainly focuses on $oxo^8G$ as the most prominent species[15,16,21–23]. On this topic we present two aspects. First, we demonstrate that oxidation of rRNA in living yeast occurs under physiological conditions, without addition of external HOCl, as evidenced by AAS signals (Fig. 1b, c). Second, we show that in these samples, abasic sites are present in higher amounts than $oxo^8G$ (Fig. 5b). Additionally, we have proven that treatment with HOCl, used as a cell-penetrating chemical tool, leads to a marked further increase in abasic sites and, at best, to a moderate increase of

oxo$^8$G, both in vitro (Fig. 3j, k) and in vivo (Fig. 5b). In in vitro oxidation experiments with higher concentrations of HOCl, oxo$^8$G disappears entirely and abasic sites increase concomitantly, strongly suggesting that oxo$^8$G is a minor and transient damage product, and subject to further oxidation. According to literature, further oxidation proceeds via several ring-opened intermediates[12–16]. Our oxidation series data revealed a sequential order of reactivity aligning with the nucleoside redox potentials. The descending order of adenosine (1.4 V) relative to guanosine (1.3 V) and 8-oxoguanosine (0.74 V) indeed argues for our scenario[20,69]. Thus, oxo$^8$G is the most easily oxidized, when present in an oligonucleotide also containing guanine, adenine, cytosine and uracil and therefore does not accumulate, a fact we observed repeatedly (Supplementary Fig. 6g, h). Rather, upon a second oxidation event in the same oligonucleotide, the oxo$^8$G is preferentially consumed over neighboring nucleobases, including guanines. Importantly, the consumption of oxo$^8$G and its lack of accumulation mean that it may be less relevant than currently perceived in the literature. Moreover, its transient character suggests that it is a suboptimal biomarker for oxidative stress, and its biological impact might be more limited than anticipated. Instead, oxidation proceeds through a number of ring-opened intermediates and yields in abasic sites, as demonstrated by LC-MS. Although some of our data even argues for a protective effect, where guanosine and oxo$^8$G prevent the oxidation of other nucleosides with a higher redox potential, the observed effects were too strongly influenced by nucleoside composition and sequence to draw general conclusions.

Meanwhile, while the RNA backbone is chemically fragile under some conditions, it is remarkably stable against HOCl oxidation yet rendered readily amenable to aniline-induced cleavage. Since oxo$^8$G is rather insensitive to aniline, the observed AAS signals must therefore originate from abasic sites, with possible minor contributions from ring-opened intermediates. Indeed, compounds such as oxazolone (rZ) and 5-carboxamido-5-formamido-2-iminohydantoin (r2Ih), characterized by the Burrows lab[51], show appreciable rates in aniline cleavage, arguing their potential contribution to overall AAS signals. While developing these lines of evidence, we performed control experiments in which the initial fragmentation by alkaline treatment was replaced by fragmentation with Mg$^{2+}$. This alteration was based on the working hypothesis that, in the absence of hydroxide anions, the glycosidic bond in the oxo$^8$G nucleoside remains intact, preventing the corresponding sites from aniline cleavage[51]. Indeed, under the altered conditions, oxo$^8$G-dependent signals were significantly reduced relative to strong signals observed for abasic sites, as tested with synthetic oligonucleotides (Fig. 4a–c). Developed into a full workflow now named OAbSeq, the method preferentially detects abasic sites, with possible minor contributions of ring-opened intermediates. Our broader findings revealed that abasic sites and oxo$^8$G occur at the same guanosine sites, that abasic sites derive from oxo$^8$G (Fig. 5), and that abasic sites are even more abundant (Fig. 3j, k and Fig. 5b). Therefore, though OAbSeq signals arise almost exclusively from abasic sites, they represent oxidative damage at guanosine sites. In addition, data from Fig. 1 reveal aniline-induced cleavage at adenosine sites, consistent with limited amounts of oxo$^8$A. At higher concentrations of HOCl, oxo$^8$A may undergo further oxidation, but obviously to a lesser degree than guanosine and oxo$^8$G. We therefore presume that a subset of adenosines is oxidized giving rise to AAS signals, which we exempted from closer examination in this report.

The application to ribosomes in vivo and in vitro served primarily to validate that AAS signals are not randomly distributed but reflect site-specific susceptibilities. Of note, HOCl-mediated oxidation hits the same positions already oxidized by ROS under physiological conditions. Beyond mechanistic validation, the ribosome data offers some perspective for applying the method to biologically generated HOCl, such as activated macrophages, where HOCl was reported to be produced at almost millimolar concentrations[70]. Since our in vivo data indicate similar RNA oxidation patterns generated by endogenous ROS and HOCl, the developed OAbSeq can likely be applied to monitor oxidation of any RNA sequence under diverse conditions, including exogenous stimuli, aging, or genetic perturbations such as RNAi or CRISPR-Cas. We have successfully explored the detection of RNA oxidation sites beyond ribosomal RNA in yeast noncoding and coding RNAs, as well as in mouse miRNA. Significantly, our finding of oxidized guanosine sites in the seed region of miRNA is in agreement with previous reports on oxo$^8$G on miRNA[9,10] but also opens the possibility that miRNA biology is influenced by abasic sites in addition to oxo$^8$G. Like AAS, OAbSeq enables transcriptome-wide mapping, allowing oxidation events to be linked to certain loci in the cell, such as mitochondria or nuclei. As a side benefit, OAbSeq also detects abasic sites arising from non-oxidative processes, potentially uncovering unknown pathways. As the biology of abasic sites in RNA remains underexplored[71], OAbSeq offers a powerful tool for future discovery.

## Methods
Additional methods are described in supplementary information.

### Quantification of free chlorine in hypochlorous acid (HOCl)
Free chlorine in HOCl solution (Thermo Fisher Scientific, Germany) was quantified according to the ISO standard protocol Part 2 "Colorimetric method using $N,N$-Diethyl−1,4-phenylenediamine (DPD), for routine control purposes" (ISO 7393-2:2017)[72]. The method is based on the formation of a pink-colored oxidation product of DPD (Sigma Aldrich, Germany) which is measured using UV spectroscopy at 515 nm. The detailed protocol is described in the supplementary information. Briefly, standard solutions for calibration purposes were prepared, including potassium iodate (KIO$_3$, Carl Roth, Germany) and potassium iodide (KI, Sigma-Aldrich, Germany) in concentrations ranging from 0.5 mg/L to 5.0 mg/L KIO$_3$. Sulfuric acid (H$_2$SO$_4$, Carl Roth, Germany) was added to a final concentration of 10 mM to oxidize KIO$_3$ and incubated for precisely 1 min at 25°C before the addition of NaOH (NaOH, VWR, Germany) to a final concentration of 20 mM. 50 μL of a DPD buffer pH 6.5 and 50 μL of a freshly prepared DPD solution were added. Standards were directly measured on a Jasco UV spectrophotometer V-650 at 515 nm. Determination of free chlorine in HOCl solution (Thermo Fisher Scientific, Germany) was performed by diluting the stock solution in ultrapure water in a final volume of 1000 μL to achieve absorbance values within the calibration range. 50 μL DPD buffer and 50 μL DPD solution were added, and the color was measured spectrophotometrically at 515 nm. The amount of free chlorine in HOCl stock solution was calculated by converting KIO$_3$ into Cl$_2$ (10.06 μg of KIO$_3$ is equivalent to 0.141 μmol Cl$_2$) according to the ISO standard protocol (ISO 7393-2:2017) cited above[72].

### HOCl oxidation reactions
Oxidation reactions were conducted in final concentrations of 50 mM phosphate buffer (potassium salt) at pH 7.4 (Carl Roth, Germany) and 5 mM MgCl$_2$ (Carl Roth, Germany). The reaction was performed at 25°C in black tubes in a final volume of 120 μL. The concentrations of HOCl (Thermo Fisher Scientific, Germany) were indicated as chlorine equivalents based on the amount of free chlorine (Cl$_2$) quantified according to the procedure described in the previous paragraph "Quantification of free chlorine in hypochlorous acid (HOCl)". Cl$_2$ concentrations ranged from 0 μM to 20 mM as indicated for the respective experiment. Two stock solutions, A and B, (60 μL each) were prepared. Solution A was a phosphate-buffered solution (50 mM pH 7.4) containing either nucleosides (100 μM), oligonucleotides (25 ng/μL) or RNA (25 ng/μL) (described in further detail in the supplementary information) and 10 mM MgCl$_2$. S. c. cells (BY4742α) were grown in standard Yeast Extract/Peptone/Dextrose (YPD) until the mid-exponential phase (0.71 OD$_{600}$). The cultured cells were subjected to

rigorous washing procedures to remove any traces of media before being resuspended in 1X PBS. The second stock solution, designated as B, consisted of a 2X concentration of HOCl (indicated as $Cl_2$) in phosphate buffer (50 mM pH 7.4). The reaction was initiated by the addition of 60 μL HOCl stock solution B to stock solution A followed by thorough vortexing, centrifugation and incubation for precisely 10 minutes at 25°C in the dark. Quenching was performed by the addition of DTT (Carl Roth, Germany) to a final concentration of 10 mM, after which the samples were vortexed and stored at -20 °C until further processing.

### Sample preparation for nucleoside quantification using LC-MS/MS

Before subjecting oligomers or RNA to LC-MS/MS, samples were digested down to nucleosides according to following conditions: RNA was incubated for 2 h at 37°C in 5 mM Tris pH 8 (Thermo Fisher Scientific, Germany) and 1 mM $MgCl_2$ using 0.6 U nuclease P1 from *P. citrinum* (Sigma-Aldrich, Germany), 0.2 U snake venom phosphodiesterase from *C. adamanteus* (Worthington Biochemicals, USA), 0.2 U alkaline phosphatase from bovine intestinal mucosa (Sigma-Aldrich, Germany), 10 U benzonase (Sigma-Aldrich, Germany), 200 ng pentostatin (Sigma-Aldrich, Germany) and 500 ng tetrahydrouridine (Merck-Millipore, Germany). 37.5 pmol nucleoside, 25 ng of binary oligoribonucleotide, and 50 ng of RNA or 38mer were used for LC-MS/MS measurements.

### Sample preparation for 5'-ribose phosphate (5'-RP) and oxo⁸GMP quantification using LC-MS/MS

RNA digestion to the monophosphate level was performed in 20 mM $NH_4OAc$ pH 5.3 (Sigma-Aldrich, Germany) and 0.2 mM $ZnCl_2$ (Sigma-Aldrich, Germany). Samples were supplemented with 0.3 U nuclease P1 from *P. citrinum* (Sigma-Aldrich, Germany) and incubated at 37°C for 1 h. 100 ng was used for 5'-RP and oxo⁸GMP quantification measurements.

### LC-MS/MS measurement of oxygenated and chlorinated nucleosides

Digested samples were measured on an Agilent 1260 Infinity (II) LC (Agilent Technologies) equipped with a diode array detector (DAD) and a Triple Quadrupole mass spectrometer (Agilent 6470). A Synergy Fusion RP column (4 μm particle size, 80 Å pore size, 250 mm length, 2 mm inner diameter) from Phenomenex was used with following settings: column temperature: 35°C and a flow rate of 0.350 mL/min using a gradient over 54 min with the following mobile phases A: 10 mM $NH_4OAc$ pH 5.8 (Thermo Fisher Scientific, Germany) using acetic acid and B: pure acetonitrile (Thermo Fisher Scientific, Germany). The gradient included the following steps: 0% B (0–10 min), 0–10% B (10–30 min), 10–40% B (30-40 min), 40–0% B (40–43 min), and a final constant composition of 0% B (43–54 min). HPLC separation was followed by a photometric measurement of nucleosides at 254 nm before entering the Triple Quadrupole mass spectrometer. Measurements were performed in the positive ion mode using Agilent MassHunter software, and modified nucleosides were monitored by dynamic multiple reaction monitoring (dynamic MRM mode). Settings used for the measurement and detection of nucleosides, including the ¹³C- and ¹⁵N-labeled guanosine, are listed in the supplementary information.

### LC-MS/MS measurement of 5'-RP and NMPs

Monophosphate digestion mixtures were measured on an Agilent 1260 Infinity (II) LC (Agilent Technologies) equipped with a diode array detector (DAD) and a Triple Quadrupole mass spectrometer (Agilent 6470). The measurements were performed using an InfinityLab Poroshell 120 EC-C18 column (Agilent Technologies), 2.7 μm particle size, 80 Å pore size, 150 mm length, 3 mm inner diameter) with the

following settings: column temperature: 15°C and a flow rate of 0.350 mL/min using a gradient over 45 min with the following mobile phases A: 25 mM $NH_4OAc$ pH 5.2 (Thermo Fisher Scientific, Germany) using acetic acid and B: pure acetonitrile (Thermo Fisher Scientific, Germany). The gradient included the following steps: 0% B (0–2 min), 0–1% B (2–2.1 min), 1% B (2.1–3 min), 1%-2% B (3–3.1 min), 2% B (3.1 – 4 min), 2–3% B (4–4.1 min), 3% B (4.1–5 min), 3–4% B (5–5.1 min), 4% B (5.1–6 min), 4–5% B (6–6.1 min), 5% B (6.1–7 min), 5–10% B (7–10 min), 10% (10–11 min), 10%-90% B (11–20 min), 90% B (20–25 min), 90–1% B (25–35 min), 1% B (35–42 min) and a final constant composition of 0% B (42–45 min). HPLC separation was followed by a photometric measurement of nucleotides at 254 nm before entering the Triple Quadrupole mass spectrometer. Measurements were performed in the positive ion mode using Agilent MassHunter software, and modified nucleotides were monitored by dynamic multiple reaction monitoring (dynamic MRM mode). Detailed MS parameters are listed in the supplementary information.

### LC-MS/MS data analysis

Quantification was performed by external calibration using standard solutions of main nucleosides, nucleotides and oxidation products (ho⁵C, ho⁵U, Cl⁵C, Cl⁵U, oxo⁸G, oxo⁸A, Cl⁸G, Cl⁸A, 5'-RP). Main nucleosides and nucleotides were quantified based on the UV signal, ranging from 1 pmol to 500 pmol, whereas chlorinated and oxygenated nucleosides were quantified in the mass spectrometer in the range of 1 fmol to 5000 fmol. The 5'-RP calibration solutions were prepared in the range from 100 fmol to 500000 fmol and quantified in the mass spectrometer. To achieve an external standard for oxo⁸GMP, the synthetic 38mer was used, which includes one oxo⁸G. The 38mer was digested to nucleotide level and amounts in the range from 1 fmol to 1000 fmol were used for external calibration. The absolute amount of each nucleoside or nucleotide was calculated as described in Kellner et al. 2014[47] by dividing the MS signal by a relative response factor (*rRFN* = slope of the respective standard calibration curve). Due to the lack of an unaffected and constant nucleoside, normalization was performed by dividing the absolute amount through the absolute amount of all nucleosides present in untreated control samples for analysis performed at the nucleoside level. Normalization of 5'-RP and oxo⁸GMP was performed to the injected amount and the number of guanosines present in the sequence. In vivo oxidized yeast RNA was normalized to an estimated total RNA proportion of 80% rRNA (total length 5475 nt)[73], 15% tRNA (76 nt)[74] and 5% mRNA (mean length 1700 nt)[75] and a guanosine content of 25%. Data was analyzed using Agilent MassHunter Qualitative Analysis 10.0 (Build 10.0.10305.0).

### ³²P- labeling of synthetic oligomer

³²P- labeling of synthetic oligonucleotides was performed using T4 Polynucleotide Kinase (PNK) 10 U/μL (Thermo Fisher Scientific, Germany) according to the manufacturer's instructions. Briefly, the reaction was performed in 100 μL including 1X commercial PNK buffer, final concentrations of 2.5 μM oligonucleotide, 2.5 μM cold ATP (Carl Roth, Germany), 1 μM ³²P-labeled ATP (Hartmann Analytic, Germany) and 0.5 U/μL PNK for 30 minutes at 37°C. To purify ³²P-labeled oligonucleotide from excess ATP, 15% denaturing PAGE was performed using a storage phosphor screen to visualize radiolabeled bands. The upper band corresponding to the oligonucleotide was excised and purified according to the standard procedure described above. ³²P-labeling experiments were performed as *n* = 1 to minimize exposure to radioactive material.

### Mapping RNA oxidative damage

The mapping method is based on the AlkAnilineSeq protocol[56]. Approximately 50-200 ng of RNA were subjected to alkaline (AAS) or/ and Mg²⁺ fragmentation (OAbSeq) protocols. RNA was subjected to alkaline fragmentation in a 50 mM bicarbonate buffer at pH 9.2 for

5 minutes at 96 °C (AAS) and to $Mg^{2+}$ fragmentation using 2 mM $MgCl_2$ in 100 mM Tris-HCl buffer (pH 8.0) for 3 minutes at 96 °C (OAbSeq). Following fragmentation, the reaction was stopped by ethanol precipitation using 3 M AcONa, pH 5.2 along with GlycoBlue™ (Invitrogen) and the mixture was snap-frozen in liquid nitrogen. After centrifugation, the pellet was washed with 80% ethanol and resuspended in nuclease-free water. The fragmented RNA was then subjected to 3′-end dephosphorylation using 5 U of Antarctic Phosphatase (NEB, USA) for 1 h at 37°C. Post-incubation, the enzyme was inactivated and RNA fragments were extracted using a phenol: chloroform (1:1) mixture and ethanol-precipitated. Following this, the RNA pellet was washed with 80% ethanol and air-dried. The dried RNA pellet was dissolved in 1 M aniline (Sigma-Aldrich) pH 4.5 and incubated for 15 min at 60°C in the dark. The reaction was stopped by ethanol precipitation as described above. The mixture was vortexed and stored overnight at -80°C before centrifugation at 13,000 x*g* at 4°C for 30 minutes. The supernatant was removed and the pellet was washed twice with 80% ethanol. The RNA pellet was air-dried and dissolved in RNAse-free water.

Library preparation was performed using the NEBNext® Small RNA Library Prep Set for Illumina® (NEB, USA) following the manufacturer's guidelines. The quality of the libraries was assessed using a high-sensitivity DNA assay on a Bioanalyzer 2100 instrument (Agilent Technologies, USA). Additionally, the libraries were quantified employing a Qubit 3.0 fluorometer (Invitrogen, USA). For sequencing, libraries were multiplexed and sequenced at a final concentration ranging between 650–750 pM on an Illumina NextSeq 2000 platform with 50 bp single-read sequencing.

Bioinformatic analysis was performed as follows: raw sequencing reads were trimmed using Trimmomatic v0.39 to remove adaptor sequence and aligned to the appropriate RNA reference sequence (synthetic 38mer oligo, *H. sapiens* tRNA$^{Asp}$, *S. cerevisiae* rRNA) using bowtie2 v2.4.2[76] in the end-to-end mode. Multiply mapped reads were excluded. Counting of 5′-reads' extremities was done using custom awk script. Normalized G Counts (NormGcounts) score was calculated as a number of aligned sequencing reads starting at a given position in RNA divided by local background signal; defined as median of AAS signals for all non-G residues in the 11 nt window (± 5 nt). Calibration curve for modification rate to NormGcounts score is non-linear and shows a higher sensitivity at low modification rates. The threshold was established at 10 times the median of normalized G counts (NormGcounts), approximately equivalent to 10 NormGcounts. A valid oxidation signal met the threshold, exhibited coverage exceeding 1000, and ensured that the random error of the NormGcount score was verified to be less than 10%, thus guaranteeing accuracy and reliability. Random errors were calculated as the square root of the number of aligned sequencing reads starting at a given position in RNA, divided by the square root of the local background signal[77]. RNA-seq analysis was performed on following replicates: all synthetic RNA ($n = 3$) except oxo$^8$G 38-mer ($n = 6$), in vitro oxidation on isolated total *S. c.* rRNA ($n = 3$), in vitro oxidation on isolated *S. c.* ribosomes ($n = 2$), in vivo oxidation on *S. c.* rRNA ($n = 3$).

### Detecting oxo$^8$G and Abasic Sites in RNA mixtures using AAS and OAb

To determine the signal strength of AAS and OAbSeq, unmodified 38mer, oxo$^8$G 38mer and rAP 38mer (see supplementary information for rAP 38mer synthesis) were mixed in varying proportions (see supplementary information), diluted to a total volume of 20 µL, and then equally divided into two PCR tubes. To each tube, either 10 µL of bicarbonate buffer or 10 µL of $Mg^{2+}$ were added and AAS and OAbSeq were performed as described above.

### Reporting summary

Further information on research design is available in the Nature Portfolio Reporting Summary linked to this article.

## Data availability

The sequencing data generated in this study have been deposited in the European Nucleotide Archive database under accession codes PRJEB73991, PRJEB84071 and PRJEB92107. The raw mass spectrometry data are provided in the Source data file. Source data are provided with this paper.

## Code availability

The code has been deposited in GitHub at https://github.com/YuriMotorin/RNAox.

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

## Acknowledgements
We thank Dr. Lilia Ayadi and Dr. Rahul Mishra for constructs containing yeast rDNA fragments, A. Marano and Dr. H. Kempf for help in mouse dissection, and Prof. Kristina Friedland and Dr. Marco Jörg for supervising mouse experiments. This project was funded by the Deutsche Forschungsgemeinschaft (DFG, German Research Foundation), TP C01 in TRR319, Project-ID 439669440 to M.H. KR was supported by LUE (Lorraine Université d'Excellence) PhD fellowship. YM was supported by ANR "OxiXlink" PRC ANR-23-CE44-0025.

## Author contributions
Y.M., M.H. conceived and supervised research; M.W., K.R., C.J.G., V.B., L.-M.K., D.G., C.K. performed the experiments; M.W., K.R., C.J.G., V.B., L.-M.K., Y.M., M.H. obtained funding for this work; M.W., K.R., V.M., Y.M., M.H. wrote the manuscript.

## Funding

## Competing interests
M.H. is a consultant for Moderna Inc. The remaining authors declare no competing interests.
