## [Transparent Peer Review file · Nature Communications]

Mapping of HOCl-oxidized RNA identifies abasic sites as major damage and oxidation product of oxo8G

Corresponding Author: Professor Mark Helm

Version 1:

Reviewer comments:

Reviewer #1

(Remarks to the Author)

I was asked to review, not the present manuscript, but rather the authors' revisions and responses to the originally submitted manuscript. I did not see the original manuscript submitted to a different Nature journal; consequently, it is a little difficult to see what changes were made. I will simply accept what the authors say in their response document.

I believe that the authors took major steps to rewrite and clarify the manuscript, which was a recurring issue with all 3 reviewers. In addition, the research team had performed a considerable number of new experiments that are now described in this version of the work. Overall, they do a good job of replying to critiques by clarifying the goals of the work, the justification for using HOCl (because it is orthogonal to endogenous ROS in yeast) and they point out that Alk-aniline-seq has been studied with many other RNA modifications. More control studies have been added.

The areas where I still see issues with the manuscript are these:

1. Aniline, as pointed out by the authors, is slow to cleave at oxo8G in RNA, so it is no surprise that oxo8G is found in low abundance. According, I don't think most researchers will accept the protocol described here as a valid sequencing protocol for oxo8G; for AP sites, yes this works, but the response is not good for oxo8G.
2. The concentration profiles show that something odd happens around 75 μ M HOCl. For most of the lesions reported in Figs. S4 and S5, there is a build-up ramping up to 75 μ M, then things decline after that. The authors interpret this as oxo8G getting further oxidized. However, it is known that HOCl reactions with nucleic acids are extremely sensitive to pH. Likely there is a change in mechanism at about 75 μ M, either due to a small shift in the buffer or something else changing in the solution. This is true for many of the products studied, not just oxo8G.
3. Because of #2, I think the conclusion that oxo8G is a transient product that is readily converted to something else that is then aniline-labile is probably not quite correct.

Overall, I think that the authors have responded well, but that there are too many lingering questions to suggest publication in Nat. Commun. at this point. Reviewer 1 states this well in saying that the results are very much in-line with what has already been published and that the present work, while of interest, does not really "move the needle." I agree with this, and also feel that the casual reader of this manuscript would be misled by the conclusions that are not yet adequately supported.

Reviewer #2

(Remarks to the Author)

The authors have addressed all my questions. The message of the manuscript is quite clear:

Under normal growth conditions oxo8G and abasic sites are at similar levels.

With oxidant treatment oxo8G shows modest increase, most appear to be converted to more oxidized form and abasic sites. oxo8G appears to serve as the sink for increased oxidants inside cells.

These results have numerous implications to our studies of oxidative stress in cells. I really like the work, the results and conclusion. It calls into question of previous results.

I have two minor suggestions:

Under modest inflammation conditions I wonder what are the main oxidative product, a prolonged low dose oxidation treatment. I wonder would oxo8G accumulate.

There are a couple high profile papers on microRNA oxo8G being functionally important. This current work calls into question of the conclusions there. Any comment from the authors?

This is quite an insightful work. I would support its publication in Nat. Communication or Nat. Chem. Biol. in this current form.

Reviewer #3

(Remarks to the Author)

The authors propose a modification to the fragmentation step of the AlkAnilineSeq (AAS) protocol, termed OAbSeq, as a tool for mapping the prevalence of abasic sites in RNA. The authors induce oxidative damage on various RNA substrates and live yeast using HOCl, and characterize the resulting oxidative products with LC-MS and ARP analysis. The authors show that guanosine (G) is the nucleobase most susceptible to oxidation, and that increasing oxidation levels cause first a depletion of G bases, then a transient increase and then depletion of 8oxoG levels, and ultimately an accumulation of abasic sites. The authors thus show that abasic sites are the predominant product and marker of guanosine oxidation under intensely oxidative conditions, rather than lesions such as 8oxoG, which are quickly further oxidized to form ring-opened guanosine derivatives and then oxidized again to form abasic sites. Based on this data, as well as their analysis of various permutations of binary oligonucleotides, the authors propose that G and 8oxoG can function as oxidative sponges that protects other nucleobases or molecules from oxidation by absorbing multiple oxidation equivalents.

The study employs a comprehensive experimental approach, including substantial and highly relevant LC-MS analysis of oxidized RNAs and their derivatives and spanning multiple model systems to validate their findings, with a primary focus on ribosomal RNA (rRNA). This focus on rRNA is strategic due to its abundance and essential cellular role, making it an ideal substrate for developing and validating their method. In vitro experiments with synthetic oligonucleotides and IVTs provide controlled environments to characterize the chemical progression of oxidative damage, while studies on isolated ribosomes offer insights into how protein-RNA interactions might influence oxidation patterns in rRNA. Their in vivo experiments in yeast cells demonstrate that HOCl can induce oxidative damage patterns in rRNA similar to those observed in vitro, but with important contextual differences that reflect the complex cellular environment. Particularly revealing is their observation that sites naturally prone to oxidation under physiological conditions are the same sites preferentially oxidized upon HOCl treatment, suggesting inherent structural or sequence-based vulnerabilities in certain rRNA regions.

This development of a precise method for sequencing and mapping abasic sites in RNA represents a significant advancement for the field of RNA damage research. The ability to site-specifically detect these lesions will enable researchers to better understand the relationship between RNA oxidation, cellular stress responses, and disease progression. Given that previous research has largely focused on 8oxoG as the primary marker for oxidative damage, this paradigm shift toward monitoring abasic sites could provide more accurate and comprehensive picture of RNA oxidation dynamics if these observations hold true for other model systems and RNA subtypes. While the current study primarily demonstrates the technique on rRNA, the method holds promise for application to other RNA species including mRNA and non-coding RNAs. Our laboratory eagerly anticipates testing this technology, as it promises to reveal previously undetectable patterns of RNA damage across various physiological and pathological conditions. The potential applications range from investigating oxidative stress in aging and neurodegenerative disorders to monitoring RNA integrity during cellular responses to environmental toxins. However, while the authors present well-reasoned arguments, there are instances where the presented data appear to not entirely support the stated conclusions concerning the ability of G and 8oxoG to protect other nucleobases from oxidation. There are also instances where the narrative structure could be improved and the data could be explained a little more clearly. Specifically, the manuscript would benefit from reorganizing for better clarity, streamlining lengthy sentences, improving figure references by stating findings before citing figures, adding more descriptive subheadings, highlighting key findings more explicitly, clarifying technical terms for broader accessibility, enhancing transitions between sections, and including summary statements for major findings. Additionally, while the focus on rRNA provided a solid foundation for method development due to its abundance, demonstrating the application of OAbSeq to other RNA species such as mRNAs or non-coding RNAs would have significantly strengthened the manuscript and provided more convincing evidence of the method's broader applicability across different experimental models and biological contexts. This expansion would help establish OAbSeq as a versatile tool for the RNA research community rather than one primarily optimized for abundant rRNA species. Overall, the experimental data presented does support the main conclusions of the paper regarding the prevalence of abasic sites as markers of oxidation damage; however, the tone and wording of some implications could be revised to avoid potentially misleading readers unfamiliar with the field.

Major Concerns:

- One of the authors' central claims is that guanosine can soak up oxidation equivalents and protect other nucleobases from oxidation, evidenced by Fig. 3A-D. However, some of the data in Fig. S7 do not seem to necessarily support this conclusion. For example, in Fig. S7A,B, the UC binary oligo has a lower concentration of C15U than the UA and UG oligos (though ho5U is more prevalent in the UC oligo), while one should think based on the authors' claims that the more easily oxidized A and Gs should protect the Us and result in lower C15U concentrations in the UA and UG oligos. As another example, in Fig. S7D,

the AG oligo has a similar or higher prevalence of 8oxoA and C18A than the AC oligo, though one would expect the AG oligo to produce fewer oxidized A derivatives if the Gs absorb more oxidative equivalents than Cs. While the overall trend of the data presented does tend to support the authors' conclusions, it might be worth addressing these discrepancies and/or tempering the assertion that G and 8oxoG can protect other nucleobases from oxidation.

- There are several instances where, while the general conclusion is justified by the results shown, the described implications are exaggerated, which can mislead readers unfamiliar with the topic.
 - o i.e. In the main text when presenting Fig 4B "clearly shows a strong, almost complete depression of the corresponding 8oxoG signal at position 16, while that of the abasic site did not diminish". There is indeed a decrease in 8oxoG signal, but it is far from almost complete.
 - o For the main text of Fig 6 "In another case, G91 in 5S rRNA, highlighted in Figure 6C is markedly oxidized only in living cells, while no or very low reactivity is noted for naked rRNA and for rRNA in isolated ribosomes." In both cases the log2 counts do not correspond to "no or very low reactivity"
 - o Several more like this through the text.
- in vivo data outside of rRNA even maintaining the same model (*S. cerevisiae*) would immediately make the method broadly applicable. As mentioned above, results and conclusions for rRNA are sound and justified overall. As a reader one keeps wanting just a little bit more, especially because the library prep was already done and while unintended by the authors the reader can be left with the feeling that non-rRNA data was just not shown. Maybe a short statement clarifying why it might have been technically challenging could solve this problem.

Minor concerns:

- Are the HOCl concentrations that end up further oxidizing 8oxoG and producing abasic sites physiologically relevant? While yeast cells don't produce HOCl, would yeast or any type of mammalian cells encounter 0.5 mM HOCl? A quick literature survey indicates they are, but either way, it would be helpful to get a clarifying statement on the manuscript to align the expectations of the readers.
- Throughout the manuscript, there are many instances where the authors introduce a technique, concept or finding but don't follow up and elaborate until later in the text, causing confusion for the reader. For example, although the authors introduce their alternative Mg²⁺ fragmentation method briefly in the introduction, when they mention it first in the results ("Importantly, the signal arising from m7G1575...omitting the alkaline treatment in AAS (Figure S1A)", it might be appropriate to describe the logic behind using this alternative fragmentation technique at this point in the results section. Furthermore, it could be worth spelling out more explicitly how Fig. S1A provides evidence "against metabolic alkylation as the cause of guanosine signals".
- Another instance where the text could be rearranged to make a more clear, step-by-step narrative is when the authors introduce ring-opened guanosine oxidation products ("we also found a number of higher oxidation products...depicted in Figure S5A") but do not reference the relevant data in Fig. S5B,C until the end of the next paragraph ("ring-opened guanosine derivatives...products of 8oxoG (Figure S5B, C)"). It could help the flow of this section to put these sentences in the same paragraph.
- One of the primary contributions of this manuscript is the OAbSeq technique, which modifies AAS to allow for improved detection of abasic sites. The manuscript would benefit from a more clear and self-contained section of text that narrates the purpose, mechanism, supporting data and benefits of OAbSeq.
- Figure 4 may benefit from an additional panel with a schematic illustrating how the reads were aligned, and how the score was derived.
- The authors argue that Figure 4C demonstrates that the contributions of the 8oxoG 38mer are negligible compared to the abasic 38mer. It might be helpful to explain why Figure 4C appears to show a multiplicative effect of the 8oxoG and abasic 38mers when mixed together. Why is it that in 4C (right), 100% 8oxoG 38mer contributes less than 50 normGcounts, 5% abasic 38mer contributes just over 50 normGcounts, but when you combine the two, we see over 400 normGcounts? Overall, it is a little difficult to interpret Figure 4C in light of the authors' claims.
- Legend in Figure 4 references a D panel. There is no D panel.
- At the start of the section describing the experiments using "oligonucleotides of binary composition", it may be helpful to point the reader towards supplemental table S5.
- Figure 5 A should also have a trendline for consistency.
- Plots with subpanels such as Fig 3G and 5B could benefit from better spatial alignment of the x axes so that it is immediately obvious what is being shown and compared. Alternatively 2 breaks on the y axis could be used to plot all the information without a subpanel.
- Fig 6B scale colors are misleading, is there a particular reason why log₂(3) should be whiteish? It gives the reader the impression that white and light purple are lower counts and unimportant when they aren't.
- Figure S11 is not mentioned in the results section and only in discussion.

We sincerely hope these comments are helpful, and are excited to try this method in the future.

Alex Chaim, PhD (together with Brandon Ho, PhD and Eli Sobel)
Assistant Professor
Department of Cell and Developmental Biology
School of Biological Sciences
University of California San Diego

Reviewer #4

(Remarks to the Author)

I co-reviewed this manuscript with one of the reviewers who provided the listed reports. This is part of the Nature Communications initiative to facilitate training in peer review and to provide appropriate recognition for Early Career

Researchers who co-review manuscripts.

Reviewer #5

(Remarks to the Author)

Version 2:

Reviewer comments:

Reviewer #1

(Remarks to the Author)

The manuscript is improved, and my questions have largely been addressed. I don't think the picture is entirely clear yet on what chemistry is actually giving rise to the abasic sites. I support publication because this report will stimulate future work.

Some minor points...

1. I think it would be better to add "HOCl" to the title in the following way, "...abasic sites as the major HOCl oxidation product..."
2. Fig. 1 cation: Surely the alkaline treatment does not "weaken the N-glycosidic". The thermodynamic bond strengths remain the same. Alkaline treatment lowers the barrier to hydrolysis of the glycosidic bond by base catalysis, hence the kinetics of bond cleavage are faster.
3. Yeast rRNA contains m7G. Would this be converted to an AP site under the conditions of study here?

Reviewer #3

(Remarks to the Author)

The authors propose a modification to the fragmentation step of the AlkAnilineSeq (AAS) protocol, termed OAbSeq, as a new tool for mapping the prevalence of abasic sites in RNA. The authors show that abasic sites rather than 8-oxoG are the predominant product of guanosine oxidation under intensely oxidative conditions, and convincingly argue that abasic sites are an important marker of oxidative damage. The revised manuscript has extended the analysis of rAP sites to ncRNA, miRNA and mRNA, which strengthens the paper. They also made several text and figure changes to clarify concepts and refine their conclusions appropriately for the data they present. Therefore, the authors have thoroughly and successfully addressed the majority of the concerns raised during the review process. The inclusion of new experimental results further strengthens the article's findings, making it suitable for publication in its current form. We are satisfied with the manuscript's revisions, and believe that the methods and findings of this work represent a strong contribution to the field.

Alex Chaim, PhD (together with Brandon Ho, PhD and Eli Sobel)
Assistant Professor
Department of Cell and Developmental Biology
School of Biological Sciences
University of California San Diego

Reviewer #4

(Remarks to the Author)

Reviewer #5

(Remarks to the Author)

Point-by-point Response to Reviewer comments

Reviewer #1 (Remarks to the Author):

I was asked to review, not the present manuscript, but rather the authors' revisions and responses to the originally submitted manuscript. I did not see the original manuscript submitted to a different Nature journal; consequently, it is a little difficult to see what changes were made. I will simply accept what the authors say in their response document.

I believe that the authors took major steps to rewrite and clarify the manuscript, which was a recurring issue with all 3 reviewers. In addition, the research team had performed a considerable number of new experiments that are now described in this version of the work. Overall, they do a good job of replying to critiques by clarifying the goals of the work, the justification for using HOCl (because it is orthogonal to endogenous ROS in yeast) and they point out that Alk-aniline-seq has been studied with many other RNA modifications. More control studies have been added.

The areas where I still see issues with the manuscript are these:

Concern 1.1. Aniline, as pointed out by the authors, is slow to cleave at oxo⁸G in RNA, so it is no surprise that oxo⁸G is found in low abundance. Accordingly, I don't think most researchers will accept the protocol described here as a valid sequencing protocol for oxo⁸G; for AP sites, yes this works, but the response is not good for oxo⁸G.

Response 1.1: We may have misphrased our intention, as we do not intend to claim that we can uniquely detect oxo⁸G. We rephrased the respective sections of the manuscript, to better emphasize that, for *in vivo* damaged RNA, we are indeed mostly mapping abasic sites. There are several key findings that we would point out in this context. First, we do NOT claim to find oxo⁸G in low abundance. Rather, our mass spec data show oxo⁸G and abasic sites occur in the same order of magnitude, with abasic sites even being slightly more abundant. This is not only VERY NEW, but in combination with the sequencing results, has important implications for the biology of oxidized guanosine sites. Second, the sequencing data show that the abasic sites occur at the same sites, where also oxo⁸G is found. Hence, at a given site, there exists a mixture of oxo⁸G and abasic sites. Third, previous methods for "mapping" of oxo⁸G have relied on RT-signatures. We have now added RT data on our model 38mer, that show that RT signatures of oxo⁸G and abasic sites are in fact very similar, strongly suggesting that previous papers have unknowingly reported signals derived from a mixture as arising exclusively from oxo⁸G, thereby ignoring the contribution of abasic sites. It follows that effects that have hitherto been ascribed to the presence of oxo⁸G might arguably have been mediated by abasic sites instead.

An additional panel showing the "no fragmentation" condition has been implemented into Figure 4C and RT signatures of synthetic 38mers including either G, oxo⁸G or an abasic site at position 16 are shown in the following Figure R1:

Figure R1. RT signature of synthetic 38mers including either G, oxo⁸G or an abasic site at position 16.

Concern 1.2. The concentration profiles show that something odd happens around 75 μ M HOCl. For most of the lesions reported in Figs. S4 and S5, there is a build-up ramping up to 75 μ M, then things decline after that. The authors interpret this as oxo⁸G getting further oxidized. However, it is known that HOCl reactions with nucleic acids are extremely sensitive to pH. Likely there is a change in mechanism at about 75 μ M, either due to a small shift in the buffer or something else changing in the solution. This is true for many of the products studied, not just oxo⁸G.

Response 1.2: Although we were and are convinced, that a 50 mM phosphate buffer cannot be overcome by 75 μ M of whatever reagent (roughly one permille of acid relative to buffering concentration), additional experiments have been performed and the results are shown in Figure S2E. Neither does the pH change upon addition of HOCl, nor does results change when using higher concentrations of phosphate buffer.

Concern 1.3. Because of #2, I think the conclusion that oxo⁸G is a transient product that is readily converted to something else that is then aniline-labile is probably not quite correct.

Response 1.3: We respectfully direct the attention of this reviewer to the series of experiments shown in Figure 3K and 4D, E. In a model 38mer containing oxo⁸G at a defined site in full stoichiometry, addition of HOCl directly leads to a decrease of oxo⁸G, meaning that oxo⁸G is oxidized faster than other guanines are converted to oxo⁸G. Simultaneously, the signal arising from aniline cleavage at that very position increases markedly, allowing the firm and, in our

opinion, irrefutable conclusion that the oxidation of oxo⁸G by HOCl leads to alkaline-labile species. We believe that we have provided sufficient experimental data to disprove this reviewer's concern. If the reviewer believes that our observations can be better explained by other uncontrolled factors, we respectfully ask for a more concrete hypothesis that can be experimentally verified or invalidated.

Concern 1.4. Overall, I think that the authors have responded well, but that there are too many lingering questions to suggest publication in Nat. Commun. at this point. Reviewer 1 states this well in saying that the results are very much in-line with what has already been published and that the present work, while of interest, does not really “move the needle.” I agree with this, and also feel that the casual reader of this manuscript would be misled by the conclusions that are not yet adequately supported.

Response 1.4: As outlined above, it is our impression that our principal message has not been well phrased and therefore not struck home. We have thus rephrased the manuscript in several places to convey the major finding: oxo⁸G and abasic sites occur at the same sites, and thus, sequencing one is sequencing the other. Among other edits, we have added a paragraph in the discussion that narrates the purpose, mechanism, supporting data and benefits of OAbSeq.

Reviewer #2 (Remarks to the Author):

The authors have addressed all my questions. The message of the manuscript is quite clear:

Under normal growth conditions oxo⁸G and abasic sites are at similar levels. With oxidant treatment oxo⁸G shows modest increase, most appear to be converted to more oxidized form and abasic sites. oxo⁸G appears to serve as the sink for increased oxidants inside cells.

These results have numerous implications to our studies of oxidative stress in cells. I really like the work, the results and conclusion. It calls into question of previous results.

I have two minor suggestions:

Concern 2.1. Under modest inflammation conditions I wonder what are the main oxidative product, a prolonged low dose oxidation treatment. I wonder would oxo⁸G accumulate.

Response 2.1: Thank you for bringing this very interesting question to our attention. As corresponding experiments are somewhat outside our current experimental setup (yeast!) we intend to address this in future experiments.

Concern 2.2. There are a couple high comment on miRNAprofile papers on microRNA oxo⁸G being functionally important. This current work calls into question of the conclusions there. Any comment from the authors?

Response 2.2: Good point, and a complicated one. Our findings do put into question the exact nature of the chemical species (in plural!) involved in the described biology, but not the biology as such.

Analysis of mouse miRNA AAS profiles (now shown in Figure 7) indeed allowed the detection of strong G-derived signals in the seed region of some abundant miRNA species. However, at this point we cannot comment on the underlying biology, which remains to be elucidated.

We are providing RT signature data as Figure R1 (Rebuttal letter p. 5), which show slight differences between abasic sites and oxo⁸G. This means that other authors may have interpreted the incorporation of dA into cDNA as an indicator for the presence of oxo⁸G, although it may as well have derived from abasic sites. According to our present findings, and as outlined above in response to reviewer #1, we would expect there to be a mixture of oxo⁸G and abasic sites, influencing the biogenesis and action of the miRNAs in question. This means the reported effects on miRNA might stem from abasic sites as well as from oxo⁸G. The authors of the miRNA papers have used antibodies directed against oxo⁸G in RNA pulldown experiments. Although the specificity of those antibodies remains to be validated in detail, this provides an argument in favor of oxo⁸G being involved, so the issue cannot be solved without further investigation, such as e.g. a pulldown of abasic site containing RNAs. We have addressed this very cautiously in the discussion.

Comment 2.3 This is quite an insightful work. I would support its publication in Nat. Communication or Nat. Chem. Biol. in this current form.

Response 2.3: Thank you!

Reviewer #3 (Remarks to the Author):

The authors propose a modification to the fragmentation step of the AlkAnilineSeq (AAS) protocol, termed OAbSeq, as a tool for mapping the prevalence of abasic sites in RNA. The authors induce oxidative damage on various RNA substrates and live yeast using HOCl, and characterize the resulting oxidative products with LC-MS and ARP analysis. The authors show that guanosine (G) is the nucleobase most susceptible to oxidation, and that increasing oxidation levels cause first a depletion of G bases, then a transient increase and then depletion of 8oxoG levels, and ultimately an accumulation of abasic sites. The authors thus show that abasic sites are the predominant product and marker of guanosine oxidation under intensely oxidative conditions, rather than lesions such as 8oxoG, which are quickly further oxidized to form ring-opened guanosine derivatives and then oxidized again to form abasic sites. Based on this data, as well as their analysis of various permutations of binary oligonucleotides, the authors propose that G and 8oxoG can function as oxidative sponges that protects other nucleobases or molecules from oxidation by absorbing multiple oxidation equivalents.

The study employs a comprehensive experimental approach, including substantial and highly relevant LC-MS analysis of oxidized RNAs and their derivatives and spanning multiple model systems to validate their findings, with a primary focus on ribosomal RNA (rRNA). This focus on rRNA is strategic due to its abundance and essential cellular role, making it an ideal substrate for

developing and validating their method. In vitro experiments with synthetic oligonucleotides and IVTs provide controlled environments to characterize the chemical progression of oxidative damage, while studies on isolated ribosomes offer insights into how protein-RNA interactions might influence oxidation patterns in rRNA. Their in vivo experiments in yeast cells demonstrate that HOCl can induce oxidative damage patterns in rRNA similar to those observed in vitro, but with important contextual differences that reflect the complex cellular environment. Particularly revealing is their observation that sites naturally prone to oxidation under physiological conditions are the same sites preferentially oxidized upon HOCl treatment, suggesting inherent structural or sequence-based vulnerabilities in certain rRNA regions.

This development of a precise method for sequencing and mapping abasic sites in RNA represents a significant advancement for the field of RNA damage research. The ability to site-specifically detect these lesions will enable researchers to better understand the relationship between RNA oxidation, cellular stress responses, and disease progression. Given that previous research has largely focused on 8oxoG as the primary marker for oxidative damage, this paradigm shift toward monitoring abasic sites could provide more accurate and comprehensive picture of RNA oxidation dynamics if these observations hold true for other model systems and RNA subtypes. While the current study primarily demonstrates the technique on rRNA, the method holds promise for application to other RNA species including mRNA and non-coding RNAs. Our laboratory eagerly anticipates testing this technology, as it promises to reveal previously undetectable patterns of RNA damage across various physiological and pathological conditions. The potential applications range from investigating oxidative stress in aging and neurodegenerative disorders to monitoring RNA integrity during cellular responses to environmental toxins.

However, while the authors present well-reasoned arguments, there are instances where the presented data appear to not entirely support the stated conclusions concerning the ability of G and 8oxoG to protect other nucleobases from oxidation. There are also instances where the narrative structure could be improved and the data could be explained a little more clearly. Specifically, the manuscript would benefit from reorganizing for better clarity, streamlining lengthy sentences, improving figure references by stating findings before citing figures, adding more descriptive subheadings, highlighting key findings more explicitly, clarifying technical terms for broader accessibility, enhancing transitions between sections, and including summary statements for major findings. Additionally, while the focus on rRNA provided a solid foundation for method development due to its abundance, demonstrating the application of OAbSeq to other RNA species such as mRNAs or non-coding RNAs would have significantly strengthened the manuscript and provided more convincing evidence of the method's broader applicability across different experimental models and biological contexts. This expansion would help establish OAbSeq as a versatile tool for the RNA research community rather than one primarily optimized for abundant rRNA species. Overall, the experimental data presented does support the main conclusions of the paper regarding the prevalence of abasic sites as markers of oxidation damage; however, the tone and wording of some implications could be revised to avoid potentially misleading readers unfamiliar with the field.

General Response to reviewer #3:

In order to demonstrate applicability of AAS and OAbSeq protocols to RNA species, different from stable rRNAs, we performed additional experiments and analysis, now showing successful detection of rAP sites in ncRNA, miR and mRNAs (see Figure 7 and Figure R1, Rebuttal letter p. 5).

Analysis of yeast *S. cerevisiae* ncRNA oxidation was performed on the already obtained datasets. Since most of our experiments have been performed using unfractionated total RNA from *S. cerevisiae*, the sequencing datasets contain not only rRNA reads, but all others, derived from non-coding and coding RNAs. Natural mRNA abundance in total RNA fraction is too low to provide reliable analysis, however, some of ncRNAs are known to be relatively well represented. Thus, in order to demonstrate applicability of our methods to such lower abundance RNAs, we performed deeper analysis of the datasets and managed to extract a sufficient number of sequencing reads for some ncRNAs. As anticipated, yeast SCR1 (RNA SRP) is sufficiently covered, as well as some snRNAs. We provide SCR1 traces in Figure S10. Indeed, treatment of RNA with HOCl generates signals at G residues in RNA (one U was also found to be reactive). As for rRNA, AAS and OAbSeq profiles for ncRNAs show substantial similarity, but some more intense signals obtained in AAS protocol might indicate the presence of oxo⁸G in addition to rAP sites in SCR1 RNA.

Extension of AAS and OAbSeq for miR analysis was demonstrated by extraction of some miR reads from datasets for mouse brain tissues (Figure 7E, F). At the depth of sequencing used for analysis of rRNAs only a few miR species are sufficiently covered, but sufficient amount of sequencing information was obtained for a dozen of most represented miR species. Many obtained AAS profiles show clear signals corresponding to G residues in miR, indicating oxidation and conversion to oxo⁸G or rAP. This validates miR oxidation analysis *in vivo*, since those samples are extracted from native mouse tissues.

Last, additional AAS analysis was performed for fractions of total yeast RNA mock and treated *in vitro* by HOCl (10 μM), as well as for total RNA extracted from *S. cerevisiae* mock cells and those treated by HOCl (1 mM). PolyA⁺ mRNA fraction was isolated from these total RNA samples and subjected to AAS analysis. With moderate sequencing depth, comparable with standard *S. cerevisiae* RNASeq transcriptomics (15 million reads/sample), only a few mRNAs got sufficient and regular coverage allowing reliable calculation of AAS scores. Despite this, the profiles obtained for biological replicates were quite comparable, and, for mRNA species analyzed, the *in vitro* and *in vivo* mRNA oxidation profiles are almost identical. We now include these data as an additional panel in Figure 7A - D.

Despite this demonstration of technical applicability of AAS and OAbSeq to analysis of RNA oxidation *in vitro* and *in vivo*, biological meaning and interpretation of such profiles is too preliminary to our opinion. Deep analysis of other biological models is certainly necessary to provide solid conclusions on the depth and possible biological consequences of such RNA oxidation. The goal of this work was to provide a validated tool for such analysis.

Major Concerns:

Concern 3.1. One of the authors' central claims is that guanosine can soak up oxidation equivalents and protect other nucleobases from oxidation, evidenced by Fig. 3A-D. However, some of the data in Fig. S7 do not seem to necessarily support this conclusion. For example, in Fig. S7A,B, the UC binary oligo has a lower concentration of C15U than the UA and UG oligos (though ho5U is more prevalent in the UC oligo), while one should think based on the authors'

claims that the more easily oxidized A and Gs should protect the Us and result in lower C15U concentrations in the UA and UG oligos. As another example, in Fig. S7D, the AG oligo has a similar or higher prevalence of 8oxoA and C18A than the AC oligo, though one would expect the AG oligo to produce fewer oxidized A derivatives if the Gs absorb more oxidative equivalents than Cs. While the overall trend of the data presented does tend to support the authors' conclusions, it might be worth addressing these discrepancies and/or tempering the assertion that G and 8oxoG can protect other nucleobases from oxidation.

Response 3.1: Your concern is indeed valid, and we agree that the results of binary oligos with the CA and CG combinations are less straightforward. At this point, we think that the redox potentials are too close for "true specificity". We performed additional oxidation experiments and LC-MS analysis of new oligomers including unreactive uridines side by side with A and G at varying positions. The analysis revealed a tendency towards differential reactivities depending on the sequence, as shown in Figure S6. As you pointed out, the overall trend of the data presented does tend to support our conclusions, but with both, the overall nucleoside composition AND sequence influencing the outcome, we must accept that a full elucidation cannot be in the scope of this paper anymore. Therefore, as you suggested, we toned down our claims.

Concern 3.2. There are several instances where, while the general conclusion is justified by the results shown, the described implications are exaggerated, which can mislead readers unfamiliar with the topic.

o i.e In the main text when presenting Fig 4B "clearly shows a strong, almost complete depression of the corresponding oxo8G signal at position 16, while that of the abasic site did not diminish". There is indeed a decrease in 8oxoG signal, but it is far from almost complete.

o For the main text of Fig 6 "In another case, G91 in 5S rRNA, highlighted in Figure 6C is markedly oxidized only in living cells, while no or very low reactivity is noted for naked rRNA and for rRNA in isolated ribosomes." In both cases the log2 counts do not correspond to "no or very low reactivity"

o Several more like this through the text.

Response 3.2: We have adjusted the strength of the statements in the above-mentioned cases and throughout the manuscript.

Concern 3.3. In vivo data outside of rRNA even maintaining the same model (*S. cerevisiae*) would immediately make the method broadly applicable. As mentioned above, results and conclusions for rRNA are sound and justified overall. As a reader one keeps wanting just a little bit more, especially because the library prep was already done and while unintended by the authors the reader can be left with the feeling that non-rRNA data was just not shown. Maybe a short statement clarifying why it might have been technically challenging could solve this problem.

Response 3.3: We now extended our analysis to other RNAs, namely ncRNA and mRNA in *S. cerevisiae* and to miRs in mouse (see details in the general response to reviewer #3 above and Figure 7). This demonstrates wide applicability of AAS/OAbSeq protocols for analysis of biologically relevant RNA oxidation.

Minor concerns:

Concern 3.4. Are the HOCl concentrations that end up further oxidizing 8oxoG and producing abasic sites physiologically relevant? While yeast cells don't produce HOCl, would yeast or any type of mammalian cells encounter 0.5 mM HOCl? A quick literature survey indicates they are, but either way, it would be helpful to get a clarifying statement on the manuscript to align the expectations of the readers.

Response 3.4: You may have overlooked (?) our statement in the discussion: ""While yeast is not known to produce HOCl, the data show that the action of HOCl in a living environment can be tracked by our method, and potentially be applied to biologically generated HOCl e.g. from macrophages, where it was reported to be produced at almost millimolar concentrations."

doi: 10.1016/S8755-9668(86)80025-4 (Weiss, 1986)

Concern 3.5. Throughout the manuscript, there are many instances where the authors introduce a technique, concept or finding but don't follow up and elaborate until later in the text, causing confusion for the reader. For example, although the authors introduce their alternative Mg²⁺ fragmentation method briefly in the introduction, when they mention it first in the results ("Importantly, the signal arising from m7G1575...omitting the alkaline treatment in AAS (Figure S1A)"), it might be appropriate to describe the logic behind using this alternative fragmentation technique at this point in the results section. Furthermore, it could be worth spelling out more explicitly how Fig. S1A provides evidence "against metabolic alkylation as the cause of guanosine signals".

Response 3.5: The corresponding section now includes more detailed explanations for OH/Mg²⁺ fragmentation and why metabolic alkylation can be ruled out.

Concern 3.6. Another instance where the text could be rearranged to make a more clear, step-by-step narrative is when the authors introduce ring-opened guanosine oxidation products ("we also found a number of higher oxidation products...depicted in Figure S5A") but do not reference the relevant data in Fig. S5B, C until the end of the next paragraph ("ring-opened guanosine derivatives...products of oxo8G (Figure S5B, C)"). It could help the flow of this section to put these sentences in the same paragraph.

Response 3.6: Although we have revised both paragraphs, we believe that moving the introduction of the ring-opened structures would rupture the narration, and the tradeoff would not lead to an overall improvement. Since the two passages are only one paragraph apart, we are confident that the reader's attention span will prove sufficient.

Concern 3.7. One of the primary contributions of this manuscript is the OAbSeq technique, which modifies AAS to allow for improved detection of abasic sites. The manuscript would benefit from a more clear and self-contained section of text that narrates the purpose, mechanism, supporting data and benefits of OAbSeq.

Response 3.7: We have added a corresponding paragraph to the discussion.

Concern 3.8. Figure 4 may benefit from an additional panel with a schematic illustrating how the reads were aligned, and how the score was derived.

Response 3.8: With due respect: the principle by which the signals are derived is shown in a panel in Figure 1, details are given in the Material and Methods section. We of course also refer to the original papers for even more details. Given that space is at a premium, we do not feel the need to supply more redundant information in this case.

Concern 3.9. The authors argue that Figure 4C demonstrates that the contributions of the 8oxoG 38mer are negligible compared to the abasic 38mer. It might be helpful to explain why Figure 4C appears to show a multiplicative effect of the 8oxoG and abasic 38mers when mixed together. Why is it that in 4C (right), 100% 8oxoG 38mer contributes less than 50 NormGcounts, 5% abasic 38mer contributes just over 50 normGcounts, but when you combine the two, we see over 400 normGcounts? Overall, it is a little difficult to interpret Figure 4C in light of the authors' claims.

Response 3.9: The library preparation protocol used in AAS (as explained in the Introduction) includes a very specific enrichment step, consisting of selective ligation of adapters uniquely to fragments possessing a 5'-phosphate extremity. Thus, only fragments resulting from aniline cleavage at the rAP site, and thus having 5'-P, are included. This makes the calibration curve non-linear, and potentially non-additive, for all scoring systems, including NormGcount score, used here. We introduced a corresponding explanation in the legend for Figure 4.

Concern 3.10. Legend in Figure 4 references a D panel. There is no D panel.

Response 3.10: Panel D has been added to Figure 4.

Concern 3.11. At the start of the section describing the experiments using "oligonucleotides of binary composition", it may be helpful to point the reader towards supplemental table S5.

Response 3.11: A pointer has been inserted on page 9, reading "(see table S5)"

Concern 3.12. Figure 5A should also have a trendline for consistency.

Response 3.12: A trendline has been added to panel Figure 5A.

Concern 3.13. Plots with subpanels such as Fig 3G and 5B could benefit from better spatial alignment of the x axes so that it is immediately obvious what is being shown and compared. Alternatively 2 breaks on the y axis could be used to plot all the information without a subpanel.

Response 3.13: Figure 3K and 5B have been changed according to the reviewer's suggestion to implement a y-axis which includes 2 breaks.

Concern 3.14. Fig 6B scale colors are misleading, is there a particular reason why $\log_2(3)$ should be whiteish? It gives the reader the impression that white and light purple are lower counts and unimportant when they aren't.

Response 3.14: Figure 6B scale colors have been changed to the classic viridis color scheme (purple to yellow).

Concern 3.15. Figure S11 is not mentioned in the results section and only in discussion.

Response 3.15: Previous Figure S11 is now Figure S6G, H and is being mentioned on page 9.

We sincerely hope these comments are helpful, and are excited to try this method in the future.

Alex Chaim, PhD (together with Brandon Ho, PhD and Eli Sobel)
Assistant Professor
Department of Cell and Developmental Biology
School of Biological Sciences
University of California San Diego

Point-by-point response

REVIEWERS' COMMENTS

Reviewer #1 (Remarks to the Author):

The manuscript is improved, and my questions have largely been addressed. I don't think the picture is entirely clear yet on what chemistry is actually giving rise to the abasic sites. I support publication because this report will stimulate future work.

Some minor points...

Comment 1.1. I think it would be better to add "HOCl" to the title in the following way, "...abasic sites as the major HOCl oxidation product..."

Response 1.1.: We have implemented HOCl in the title. Given that there is a limitation on the number of words in the title, it was changed it to: "Mapping of HOCl-oxidized RNA identifies abasic sites as major damage and oxidation product of oxo8G"

Comment 1.2. Fig. 1 caption: Surely the alkaline treatment does not "weaken the N-glycosidic". The thermodynamic bond strengths remain the same. Alkaline treatment lowers the barrier to hydrolysis of the glycosidic bond by base catalysis, hence the kinetics of bond cleavage are faster.

Response 1.2.: Changing the pH will change the thermodynamic parameters, if not enthalpy then free enthalpy. However, we have changed the caption to avoid this by writing "render it more prone to cleavage".

Comment 1.3. Yeast rRNA contains m⁷G. Would this be converted to an AP site under the conditions of study here?

Response 1.3: It is likely that m⁷G is converted, but we do not have experimental proof at this time. Since it does not pertain to the outcome, we prefer not to treat this side issue in the manuscript.

Reviewer #3 (Remarks to the Author):

The authors propose a modification to the fragmentation step of the AlkAnilineSeq (AAS) protocol, termed OAbSeq, as a new tool for mapping the prevalence of abasic sites in RNA. The authors show that abasic sites rather than 8-oxoG are the predominant product of guanosine oxidation under intensely oxidative conditions, and convincingly argue that abasic sites are an important marker of oxidative damage. The revised manuscript has extended the analysis of rAP sites to ncRNA, miRNA and mRNA, which strengthens the paper. They also made several text and figure changes to clarify concepts and refine their conclusions appropriately for the data they present. Therefore, the authors have thoroughly and successfully addressed the majority of the concerns raised during the review process. The inclusion of new experimental results further strengthens the article's findings, making it suitable for publication in its current form. We are satisfied with the manuscript's revisions,

28 August 2025

and believe that the methods and findings of this work represent a strong contribution to the field.

Response 3: We would like to thank the reviewer for his thoughtful and encouraging assessment of our work. We are pleased that our additional work and the supplementary analyses enhanced the quality of the manuscript.